# Multiple timescales of sensory-evidence accumulation across the dorsal cortex

**Lucas Pinto[1,2], David W Tank[2]\*[†], Carlos D Brody[2]\*[†]**

[1]Department of Neuroscience, Northwestern University, Chicago, United States;
[2]Princeton Neuroscience Institute, Princeton University, Princeton, United States

**Abstract** Cortical areas seem to form a hierarchy of intrinsic timescales, but the relevance of this organization for cognitive behavior remains unknown. In particular, decisions requiring the gradual accrual of sensory evidence over time recruit widespread areas across this hierarchy. Here, we tested the hypothesis that this recruitment is related to the intrinsic integration timescales of these widespread areas. We trained mice to accumulate evidence over seconds while navigating in virtual reality and optogenetically silenced the activity of many cortical areas during different brief trial epochs. We found that the inactivation of all tested areas affected the evidence-accumulation computation. Specifically, we observed distinct changes in the weighting of sensory evidence occurring during and before silencing, such that frontal inactivations led to stronger deficits on long timescales than posterior cortical ones. Inactivation of a subset of frontal areas also led to moderate effects on behavioral processes beyond evidence accumulation. Moreover, large-scale cortical $Ca^{2+}$ activity during task performance displayed different temporal integration windows. Our findings suggest that the intrinsic timescale hierarchy of distributed cortical areas is an important component of evidence-accumulation mechanisms.

**\*For correspondence:**
dwtank@princeton.edu (DWT);
brody@princeton.edu (CDB)

[†]These authors contributed equally to this work

**Competing interest:** The authors declare that no competing interests exist.

## Editor's evaluation

Pinto and colleagues used brief optogenetic silencing to study the contributions of different cortical areas in an evidence accumulation task in mice. The authors show that silencing of frontal regions affected evidence accumulation on a longer timescale than that of posterior regions, providing evidence indicating the relation between cortical functions and intrinsic timescales.

## Introduction

The cerebral cortex of both rodents and primates appears to be organized in a hierarchy of intrinsic integration timescales, whereby frontal areas integrate input over longer time windows than sensory areas (*Cavanagh et al., 2020*; *Chaudhuri et al., 2015*; *Gao et al., 2020*; *Hasson et al., 2008*; *Ito et al., 2020*; *Kiebel et al., 2008*; *Murray et al., 2014*; *Runyan et al., 2017*; *Soltani et al., 2021*; *Spitmaan et al., 2020*). Although this idea has received increasing attention, the role of such timescale hierarchy in cognitive behavior remains unclear.

In particular, the decisions we make in our daily lives often unfold over time as we deliberate between competing choices. This raises the possibility that decisions co-opt the cortical timescale hierarchy such that different cortical areas integrate decision-related information on distinct timescales. A commonly studied type of time-extended decision-making happens under perceptual uncertainty, which requires the gradual accrual of sensory evidence. This involves remembering a running tally of evidence for or against a decision, updating that tally when new evidence becomes available, and making a choice based on the predominant evidence (*Bogacz et al., 2006*; *Brody and Hanks, 2016*; *Brunton et al., 2013*; *Carandini and Churchland, 2013*; *Gold and Shadlen, 2007*; *Morcos*

*and Harvey, 2016*; *Newsome et al., 1989*; *Odoemene et al., 2018*; *Stine et al., 2020*; *Sun and Landy, 2016*; *Tsetsos et al., 2012*; *Waskom and Kiani, 2018*). Neural correlates of decisions relying on evidence accumulation have been found in a number of cortical and subcortical structures, in both primates and rodents (*Brincat et al., 2018*; *Ding and Gold, 2010*; *Erlich et al., 2015*; *Hanks et al., 2015*; *Horwitz and Newsome, 1999*; *Kim and Shadlen, 1999*; *Koay et al., 2020*; *Krueger et al., 2017*; *Murphy et al., 2021*; *Orsolic et al., 2021*; *Scott et al., 2017*; *Shadlen and Newsome, 2001*; *Wilming et al., 2020*; *Yartsev et al., 2018*). Likewise, we have previously shown that, when mice must accumulate evidence over several seconds to make a navigational decision, the inactivation of widespread dorsal cortical areas leads to behavioral deficits, and that these areas encode multiple behavioral variables, including evidence (*Pinto et al., 2019*). However, we do not understand which aspects of these decisions lead to such widespread recruitment of brain structures.

Here, we hypothesized that the pattern of widespread recruitment of cortical areas during prolonged evidence accumulation can be in part explained by their underlying timescale hierarchy. To test this, we trained mice to accumulate evidence over seconds toward navigational decisions and used brief optogenetic inactivation of single or combined cortical areas, restricted to one of six epochs of the behavioral trials. We show that the inactivation of widespread areas in the dorsal cortex strongly affects the processing and memory of sensory evidence, and that inactivating a subset of frontal areas also results in prospective behavioral deficits. Further, the inactivation of different areas affects accumulation over distinct timescales, such that, to an approximation, frontal areas contribute to evidence memory over longer temporal windows than posterior areas. In agreement with this, we show that cortical activity during the accumulation task displays a gradient of intrinsic timescales, which are longer in frontal areas. Our findings thus suggest the existing cortical hierarchy of temporal integration windows is important for evidence-accumulation computations.

## Results

### Brief inactivation of different cortical areas leads to accumulation deficits on distinct timescales

We trained mice to accumulate evidence over relatively long timescales while navigating in VR (*Figure 1A*; *Pinto et al., 2018*). The mice navigated a 3-m long virtual T-maze and during the first 2 m (~4 s) they encountered salient objects, or towers, along the walls on either side, and after a delay of 1 m (~2 s) turned into the arm corresponding to the highest perceived tower count. The towers were visible for 200 ms and appeared at different positions in each trial, obeying spatial Poisson processes of different underlying rates on the rewarded and non-rewarded sides. Compatible with our previous reports (*Koay et al., 2020*; *Pinto et al., 2018*), task performance was modulated by the difference in tower counts between the right and left sides (*Figure 1B*, n=28). Crucially, beyond allowing us to probe sensitivity to sensory evidence, the task design decorrelated the position of individual towers from the animals' position in the maze across trials. This allowed us to build a logistic regression model that used the net sensory evidence (Δ towers, or #R – #L) from each of four equally spaced bins from the cue region to predict the choice the mice made. In other words, we inferred the weight of sensory evidence from different positions in the maze on the final decision. While individual mice showed different evidence-weighting profiles, fitting the model on aggregate data yielded a flat evidence-weighting curve (*Figure 1C*, n=108,940 trials), indicating that on average the mice weighted evidence equally from throughout the maze (*Pinto et al., 2018*).

Our previous results have shown that cortical contributions to the performance of this task are widespread (*Pinto et al., 2019*), but our whole-trial inactivation did not allow us to tease apart the nature of the contributions of different areas. Here, we addressed this by using sub-trial optogenetic inactivations to ask how different dorsal cortical regions contribute to the temporal weighting of sensory evidence in order to make a perceptual decision. To do this we cleared the intact skull of mice expressing Channelrhodopsin-2 (ChR2) in inhibitory interneurons (VGAT-ChR2-EYFP, n=28) and used a scanning laser system to bilaterally silence different cortical regions, by activating inhibitory cells (*Figure 1D*; *Guo et al., 2014*; *Pinto et al., 2019*). We targeted seven different areas – primary visual cortex (V1), medial secondary visual cortex (mV2, roughly corresponding to area AM), posterior parietal cortex (PPC), retrosplenial cortex (RSC), the posteromedial portion of the premotor cortex (mM2), the anterior portion of the premotor cortex (aM2), and the primary motor cortex (M1) – as well as two

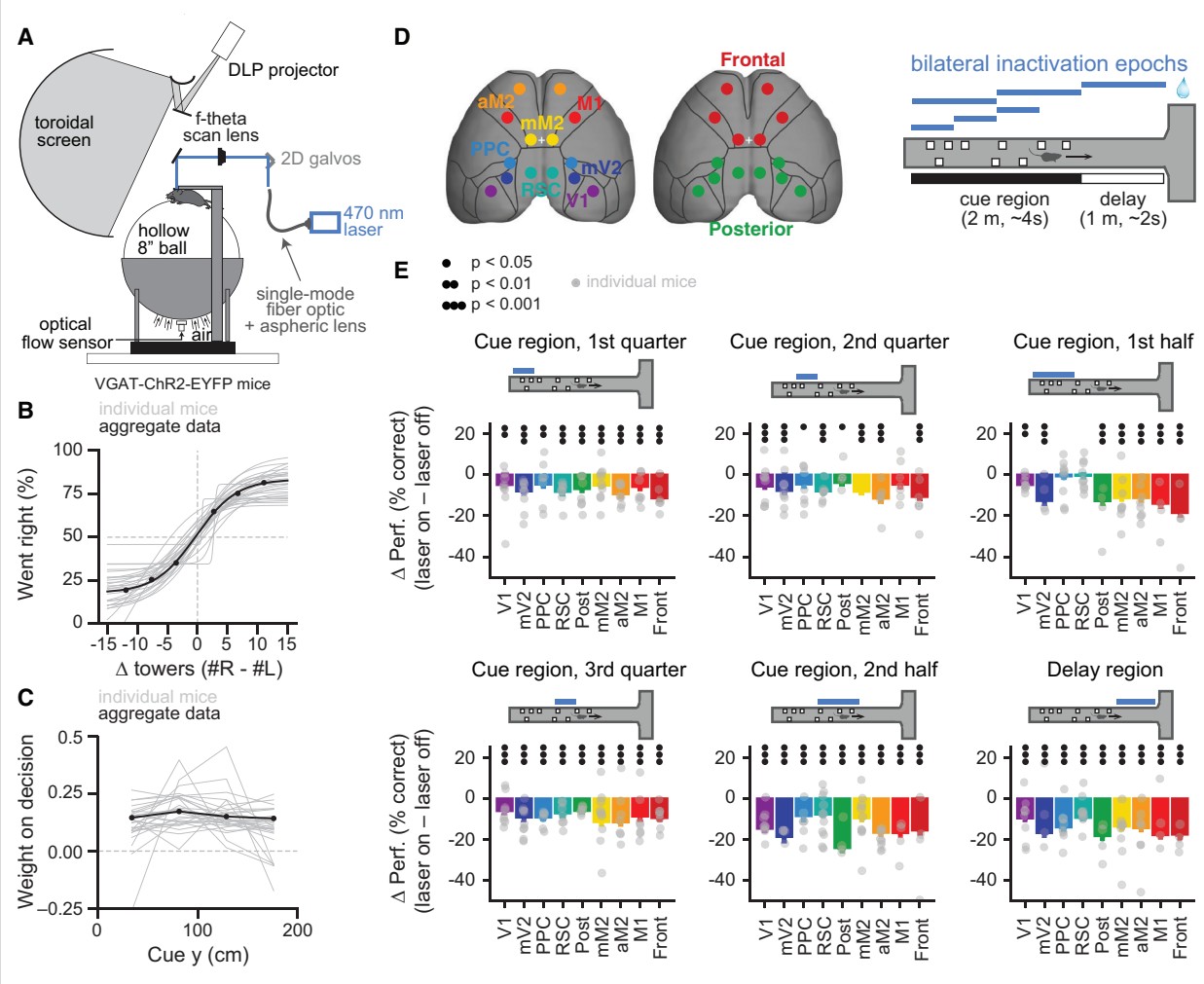

**Figure 1.** Temporally specific inactivation of multiple dorsal cortical regions during performance of a virtual reality-based evidence-accumulation task. (**A**) Schematics of the experimental setup. (**B**) Psychometric functions for control trials, showing the probability of right-side choice as a function of the strength of right sensory evidence, Δ towers (#R – #L). Thin gray lines: best fitting psychometric functions for each individual mouse (n=28). Black circles: aggregate data (n=108,940 trials), black line: fit to aggregate data, error bars: binomial confidence intervals. (**C**) Logistic regression curves for the weight of sensory evidence from four equally spaced bins on the final decision, from control trials. Thin gray lines: individual animals, thick black line: aggregate data, error bars: ± SD from 200 bootstrapping iterations. (**D**) Experimental design. We bilaterally inactivated seven dorsal cortical areas, alone or in combination, yielding a total of nine area sets, while mice performed the accumulating towers task. Bilateral inactivation happened during one of six regions in the maze spanning different parts of the cue region or delay. We thus tested a total of 54 area-epoch combinations. (**E**) Effects of sub-trial inactivation on overall performance during all 54 area-epoch combinations. Each panel shows inactivation-induced change in overall % correct performance for each inactivation epoch, for data pooled across mice. Error bars: SD across 10,000 bootstrapping iterations. Black circles indicate significance according to the captions on the leftmost panel. Light gray circles indicate data from individual mice (n=4–11, see *Figure 1—source data 1* for details about n per condition).

The online version of this article includes the following source data and figure supplement(s) for figure 1:

**Source data 1.** Numbers of mice, sessions, and trials for each of the 54 experimental conditions.

**Source data 2.** Source data for plots in *Figure 1*.

**Figure supplement 1.** Effects of sub-trial inactivation on psychometric functions during all 54 area-epoch combinations.

**Figure supplement 2.** Effects of sub-trial inactivation on evidence-weighting functions during all 54 area-epoch combinations.

**Figure supplement 3.** Inactivation of cortical areas moderately increases running speeds in some inactivation conditions.

combinations of these individual areas, namely posterior cortex (V1, mV2, PPC, and RSC) and frontal cortex (mM2, aM2, and M1). Cortical silencing occurred in one of six trial epochs: first, second , or third quarter of the cue region (0–50 cm, 50–100 cm, or 100–150 cm, respectively), first or second half of the cue region (0–100 cm or 100–200 cm, respectively), or delay region (200–300 cm). We tested all 54 possible area-epoch combinations (*Figure 1—source data 1*). This large number of experimental conditions allowed us to assess how the inactivation of different areas affects different aspects of the decision-making process.

Compatible with our previous whole-trial inactivation experiments (*Pinto et al., 2019*), we found that the inactivation of all tested cortical areas significantly affected behavioral performance, though to varying degrees (*Figure 1E*, *Figure 1—figure supplement 1*). Furthermore, we observed a variety of effect profiles across regions and inactivation epochs, as assessed by the difference between the evidence-weighting curves separately calculated for 'laser off' and 'laser on' trials (*Figure 1—figure supplement 2*). Although our previous measurements indicate inactivation spreads of at least 2 mm (*Pinto et al., 2019*), we observed different effects even comparing regions that were in close physical proximity (e.g. V1 and mV2). Additionally, all tested areas had significant effects in at least a subset of conditions (*Figure 1—figure supplement 2*, p<0.05, bootstrapping). Finally, in agreement with our previous results (*Pinto et al., 2019*), inactivation resulted in minor changes in running speed in a subset of conditions (average overall increase of ~8%, *Figure 1—figure supplement 3*). Importantly, we have previously shown that these effects are specific to mice expressing ChR2, ruling out a non-specific light effect (*Pinto et al., 2019*).

We next assessed directly whether the inactivation of different cortical areas led to changes in how much the mice based their final decision on evidence from different times in the trial with respect to inactivation. We reasoned that changes in the weighting of sensory evidence occurring before laser onset would primarily reflect effects on the memory of past evidence, while changes in evidence occurring while the laser was on would reflect disruption of processing and/or very short-term memory of the evidence. Finally, changes in evidence weighting following laser offset would potentially indicate effects on processes beyond accumulation per se, such as commitment to a decision. For example, a perturbation that caused a premature commitment to a decision would lead to towers that appeared subsequent to the perturbation having no weight on the animal's choice. Although our inactivation epochs were defined in terms of spatial position within the maze, small variations in running speed across trials, along with the moderate increases in running speed during inactivation, could have introduced confounds in the analysis of evidence as a function of maze location (*Figure 1—figure supplement 2*). Thus, we repeated the analysis of *Figure 1C* but now with logistic regression models, built to describe inactivation effects for each area, in which net sensory evidence was binned in time instead of space. Further, to account for the inter-animal variability we observed, we used a mixed-effects logistic regression approach, with mice as random effects (see Materials and methods for details), thus allowing each mouse to contribute its own source of variability to overall side bias and sensitivity to evidence at each time point, with or without the inactivations. We first fit these models separately to inactivation epochs occurring in the early or late parts of the cue region, or in the delay (y≤100 cm, 100<y≤200 cm, y>200 cm, respectively). We again observed a variety of effect patterns, with similar overall laser-induced changes in evidence weighting across epochs for some but not all tested areas (*Figure 2—figure supplement 1*). Such differences across epochs could reflect dynamic computational contributions of a given area across a behavioral trial. However, an important confound is the fact that we were not able to use the same mice across all experiments due to the large number of conditions (*Figure 1—source data 1*), such that epoch differences (where epoch is defined as time period relative to trial start) could also simply reflect variability across subjects. To address this, for each area we combined all inactivation epochs in the same model, adding them as additional random effects, thus allowing for the possibility that inactivation of each brain region at each epoch would contribute its own source of variability to side bias; different biases from mice perturbed at different epochs would then be absorbed by this random-effects parameter. We then aligned the timing of evidence pulses to laser onset and offset within the same models, as opposed to aligning with respect to trial start. This alignment combined data from mice inactivated at different epochs together, further ameliorating potential confounds from any mouse × epoch-specific differences. Each fixed-effects data point in figures below (*Figures 2 and 3*, solid colors) thus reflects tendencies common across mice, not individual mouse effects; the latter are shown as the random effects (faded colors). This approach allowed

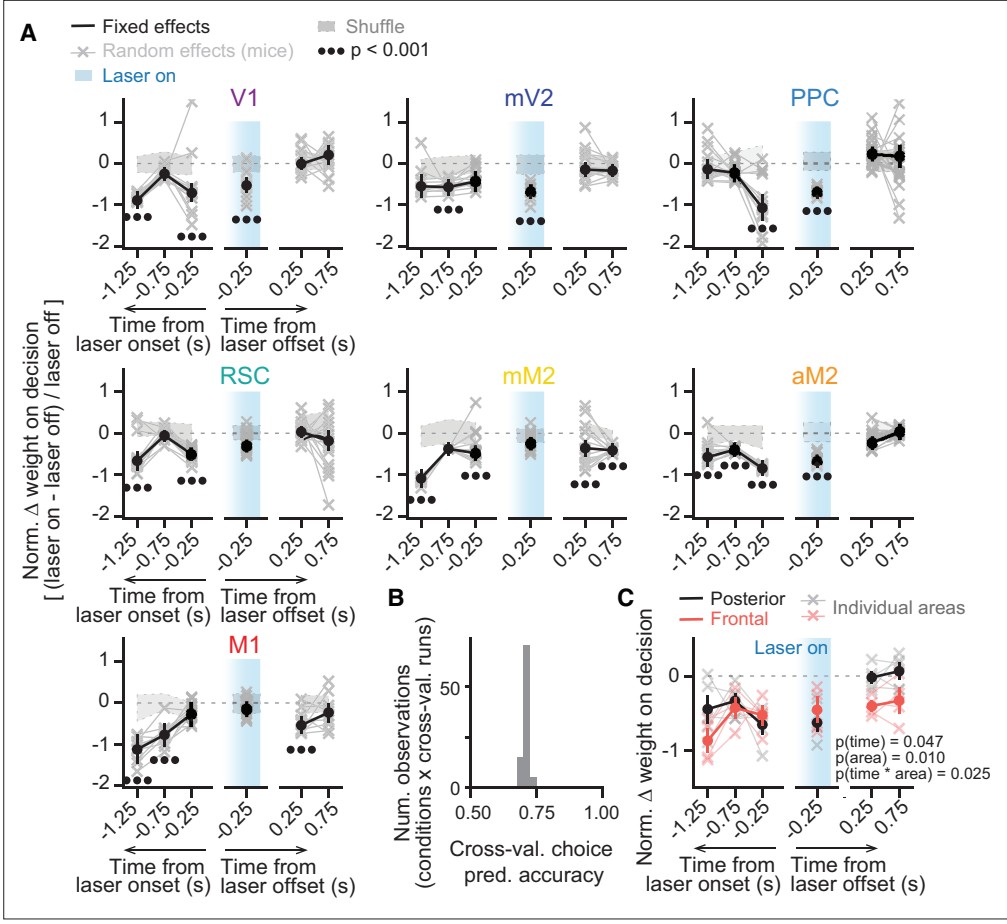

**Figure 2.** Inactivating different cortical areas leads to evidence-accumulation deficits on distinct timescales.
(**A**) Results from mixed-effects logistic regression models fit to inactivation data from different areas, combined across mice and inactivation epochs, with 10-fold cross-validation. For each area, we plot normalized evidence weights for inactivation trials, such that 0 means no difference from control, and –1 indicates complete lack of evidence weight on decision. Net evidence (#R – #L towers) was binned in time (0.5 s time bins) and aligned to either laser onset or offset. Coefficients were extracted from the model with highest cross-validated accuracy. Thin gray lines and crosses, mouse random effects (primary visual cortex [V1]: n=15, medial secondary visual cortex [mV2]: n=14, posterior parietal cortex [PPC]: n=20, retrosplenial cortex [RSC]: n=21, posteromedial portion of the premotor cortex [mM2]: n=15, anterior portion of the premotor cortex [aM2]: n=15, primary motor cortex [M1]: n=11; note that we omitted 10/666 random effect outliers outside the 1st–99th percentile range for clarity, but they were still considered in the analysis). The zero-mean random effects were added to the fixed effects for display only. Error bars, ± SEM on coefficient estimates. Black circles below the data points indicate statistical significance from t-tests, with false discovery rate correction. We also imposed an additional significance constraint that the coefficient ± SEM does not overlap with ± SD intervals for coefficients for models fit to shuffled data (gray shaded areas, see Materials and methods for details). (**B**) Distribution of model prediction accuracy, defined as the proportion of correctly predicted choices in 10% of the data not used to fit the model (n=9 areas × 10 cross-validation runs). (**C**) Comparison of the inactivation effects between areas in the posterior (V1, mV2, PPC, RSC) and frontal cortex (mM2, aM2, M1). Thin lines with crosses, individual areas. Thick lines and error bars, mean ± SEM across areas. p-values are from a two-way ANOVA with repeated measures, with time bins and area group as factors.

The online version of this article includes the following source data and figure supplement(s) for figure 2:

**Source data 1.** Source data for plots in *Figure 2*.

**Figure supplement 1.** Effects of inactivation on sensory-evidence weights across trial time appear to depend on when the inactivation occurred, for a subset of areas.

**Figure supplement 2.** Sensory-evidence weights across trial time plotted separately for control and laser trials.

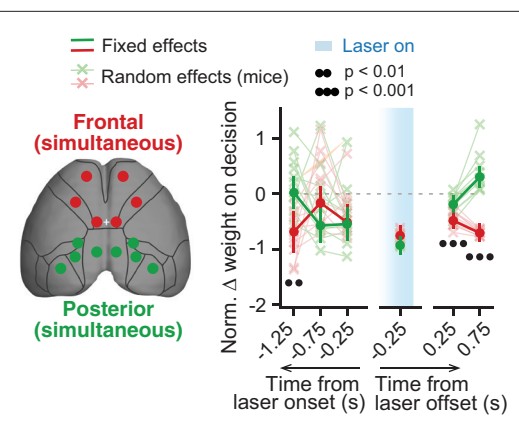

**Figure 3.** Simultaneous inactivation of frontal or posterior cortical areas confirms distinct contributions to sensory-evidence-based decisions. Comparison of the effect of simultaneous posterior or frontal cortical inactivation on the use of sensory evidence as recovered from the mixed-effects logistic regression models combined across mice and inactivation epochs, with 10-fold cross-validation. For each area, we plot normalized evidence weights for inactivation trials as in **Figure 2**. Thin gray lines and crosses, mouse random effects (posterior: n=11, frontal: n=11). Error bars, ± SEM on coefficient estimates. Black circles below the data points indicate p-values from z-tests on the coefficients, with false discovery rate correction (captions on top, see Materials and methods for details).

The online version of this article includes the following source data and figure supplement(s) for figure 3:

**Source data 1.** Source data for plots in **Figure 3**.

**Figure supplement 1.** Sensory-evidence information from different areas is combined unevenly.

us to extract the common underlying patterns of inactivation effects on the use of sensory evidence toward choice, while simultaneously accounting for inter-subject and inter-condition variability. These models confirmed that, in control conditions, evidence use is fairly constant across time (**Figure 2—figure supplement 2**), allowing us to compare the inactivation-induced deficits across time points in the trials (**Figure 2A**). Overall, these models accurately predicted ~70% of behavioral choices in trials not used to fit the data (**Figure 2B**).

This modeling approach revealed that inactivation of different areas led to deficits in the use of sensory evidence from distinct time points within the behavioral trials. For instance, PPC inactivation only led to significant decreases in the use of sensory evidence occurring during inactivation or immediately preceding it (**Figure 2A**, p<0.001, t-test), indicating a role in processing and memory of very recent sensory evidence (≤0.5 s). Similarly, we observed deficits in the use of sensory evidence during the inactivation period for all posterior areas (p<0.001, t-test), except for RSC (p>0.05). However, their role went beyond pure sensory processing, as their inactivation decreased the use of sensory evidence occurring prior to laser onset, albeit on different time scales. Inactivation of mV2 only affected evidence memory on intermediate timescales (0.5–1.0 s, p<0.001, t-test). Conversely, V1 and RSC inactivation led to non-monotonic changes in evidence memory, affecting recent (≤0.5 s, p<0.001) and long-past evidence (1.0–1.5 s, p<0.001), but not evidence occurring in between (0.5–1.0 s, p>0.05). Lastly, for all posterior cortical areas, significant changes in the evidence-weighting curves happened exclusively for evidence concomitant to or preceding laser onset, indicating that the manipulations primarily affected the processing and/or memory of the evidence, i.e., the accumulation process itself.

The inactivation of all tested frontal areas also led to deficits in evidence memory, but with different temporal profiles. Notably, inactivation of all three frontal areas (M1, mM2, aM2) led to profound deficits in the use of long-past evidence (1.0–1.5 s, p<0.001, t-test), but only aM2 inactivation led to changes in the use of evidence occurring while the laser was on (p<0.001). Interestingly, inactivation of mM2 and M1 also led to significant changes in the use of sensory evidence occurring after laser offset (p<0.001), although to a lesser extent than pre-laser evidence (**Figure 2A**). This could indicate that, in addition to evidence accumulation, these frontal areas also have a role in other decision processes, such as post-stimulus categorization (**Hanks et al., 2015**) or commitment to a decision. In this case, silencing these regions could result in a premature decision. However, the possibility remains that these effects are related to lingering effects of inactivation on population dynamics in frontal regions, which we have found to evolve on slower timescales (see below). Although we have previously verified in an identical preparation that our laser parameters lead to near-immediate recovery of pre-laser firing rates of single units, with little to no rebound (**Pinto et al., 2019**), these measurements were not done during the task, such that we cannot completely rule out this possibility.

Thus, while we observed diverse temporal profiles of evidence-weighting deficits resulting from the inactivation of different areas of the dorsal cortex, to an approximation they could be broadly

divided according to whether they belonged to the frontal or posterior cortex. Indeed, frontal and posterior areas differed significantly in terms of the magnitude and time course of evidence-weighting deficits induced by their inactivation (*Figure 2C*, two-way repeated measure ANOVA with factors time bin and area group; $F[time]_{5,15} = 3.09$, p[time]=0.047, $F[area]_{1,3} = 33.93$, p[area]=0.010, $F[interaction]_{5,15} = 3.60$, p[interaction]=0.025).

To further explore the different contributions of posterior and frontal cortical areas to the decision-making process, we next analyzed the effect of inactivating these two groups of areas simultaneously, using the same mixed-effects modeling approach as above. Compatible with our previous analysis, we found significant differences in how these two manipulations impacted the use of sensory evidence (*Figure 3*). In particular, compared to posterior areas, frontal inactivation resulted in a significantly larger decrease in the use of sensory evidence occurring long before laser onset (1.0–1.5 s, p=0.006, z-test). Moreover, it led to decreases in the use of sensory evidence occurring after inactivation (p<0.001, z-test).

Finally, we wondered whether evidence information from different areas is evenly combined, at least from a behavioral standpoint. To do this, we compared the effects of simultaneously inactivating all frontal or posterior areas to that expected by an even combination of the effects of inactivating areas individually (i.e. their average). Both posterior and frontal significantly deviated from the even-combination prediction (*Figure 3—figure supplement 1*, p<0.05, z-test). This could suggest that signals from the different dorsal cortical areas are combined with different weights toward a final decision.

## A hierarchy of timescales in large-scale cortical activity during evidence accumulation

Our inactivation results thus suggest that different regions of the dorsal cortex contribute to distinct aspects of evidence-accumulation-based decisions. In particular, while all areas we tested appear to have a role in evidence accumulation, they do so on distinct timescales. This is reminiscent of the findings that cortical areas display a hierarchy of intrinsic timescales, such that primary sensory areas tend to integrate over shorter time windows than frontal and other association areas (*Chaudhuri et al., 2015*; *Hasson et al., 2008*; *Murray et al., 2014*; *Runyan et al., 2017*). While these are thought to arise in part from intrinsic cellular and circuit properties such as channel and receptor expression, amount of recurrent connectivity, and relative proportions of inhibitory interneuron subtypes (*Chaudhuri et al., 2015*; *Duarte et al., 2017*; *Fulcher et al., 2019*; *Gao et al., 2020*; *Wang, 2020*), they appear to be modulated by task demands (*Gao et al., 2020*; *Ito et al., 2020*; *Zeraati et al., 2021*). Thus, to confirm whether this timescale hierarchy exists in the mouse cortex during performance of the accumulating-towers task, we reanalyzed previously published data consisting of mesoscale wide-field $Ca^{2+}$ imaging of the dorsal cortex through the intact cleared skull of mice expressing the $Ca^{2+}$ indicator GCaMP6f in excitatory neurons (*Figure 4A*, Emx1-Ai93 triple transgenics, n=6, 25 sessions) (*Pinto et al., 2019*). To do this, we enhanced our previous linear encoding model of the average activity of anatomically defined regions of interest (ROIs) (*Pinto et al., 2019*) by including two sets of predictors in addition to task events. First, for each ROI we added the zero-lag activity of other simultaneously imaged ROIs as coupling predictors, similar to previous work (*Pillow et al., 2008*; *Runyan et al., 2017*; *Figure 4—figure supplement 1*). Crucially, we also included auto-regressive predictors to capture intrinsic activity autocorrelations that are not locked to behavioral events. In other words, this approach allowed us to estimate within-task autocorrelations while separately accounting for task-induced temporal structure in cortical dynamics (*Spitmaan et al., 2020*). Adding these new sets of predictors resulted in a large and significant increase in cross-validated model accuracy, as measured by the linear correlation coefficient between the model predictions and a test dataset not used to fit the model (*Figure 4B and C*; ~0.95 vs. ~0.3, $F_{model\ (6,2,12)} = 1994.85$, p=6.2 × 10$^{-13}$, two-way ANOVA with repeated measures).

We first wondered whether different timescales would be reflected in model coefficients related to sensory evidence. Interestingly, however, we did not observe any significant differences across areas in the time course of coefficients for contralateral tower stimuli or cumulative sensory evidence (*Figure 4—figure supplement 1*). Thus, we next focused our analysis on the auto-regressive coefficients of the model. We observed that across animals the rate of decay of these coefficients over lags slowed systematically from visual to premotor areas, with intermediate values for M1, PPC, and RSC

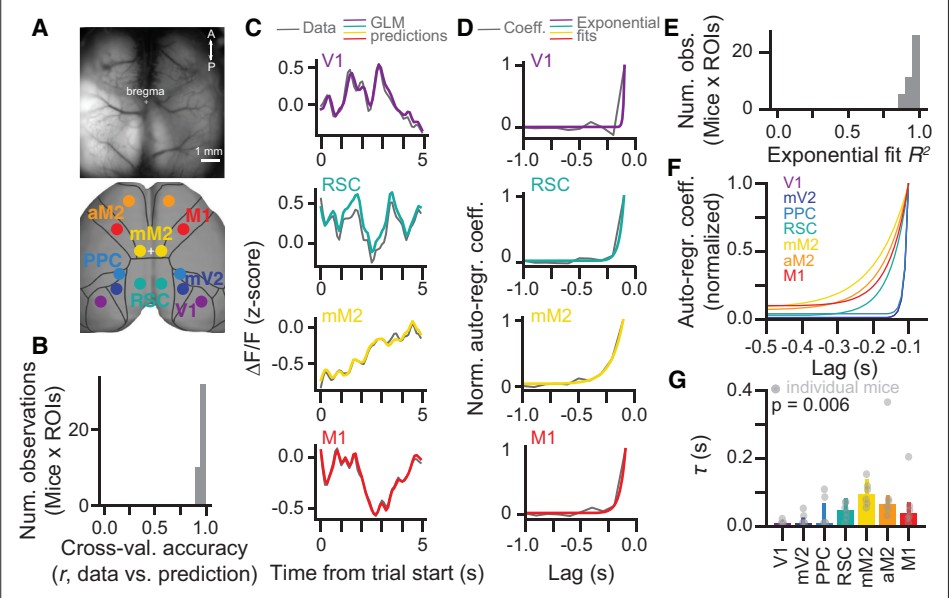

**Figure 4.** A hierarchy of activity timescales during evidence accumulation. (**A**) Top: Example widefield imaging field of view showing GCaMP6f fluorescence across the dorsal cortex. Bottom: Approximate correspondence between the field of view and regions of interest (ROIs) defined from the Allen Brain Atlas, ccv3. (**B**) Distribution of cross-validated accuracies across mice (n=6, sessions for each mouse are averaged) and ROIs (n=7, averaged across hemispheres). (**C**) Example of actual ΔF/F (gray) and model predictions (colored lines) for the first 5 s of the same held-out single trial, and four simultaneously imaged ROIs. Traces are convolved with a 1-SD Gaussian kernel for display only. (**D**) Auto-regressive model coefficients as a function of time lags for an example imaging session and four example ROIs. Gray, coefficient values. Colored lines, best fitting exponential decay functions. (**E**) Distribution of $R^2$ values for the exponential fits across mice (n=6, sessions for each mouse are averaged) and ROIs (n=7, averaged across hemispheres). (**F**) Exponential decay functions for all seven cortical areas, fitted to the average across mice (n=6). (**G**) Median time constants extracted from the exponential decay fits, for each area. Error bars, interquartile range across mice (n=6). p-value is from a one-way ANOVA with repeated measures with ROIs as factors. Light gray circles indicate data from individual mice.

The online version of this article includes the following source data and figure supplement(s) for figure 4:

**Source data 1.** Source data for plots in *Figure 4*.

**Figure supplement 1.** Time course of sensory-evidence predictors in the encoding model of widefield Ca²⁺ dynamics does not significantly differ between cortical areas.

(*Figure 4D*). To quantify this, we fitted exponential decay functions to the auto-regressive coefficients averaged across hemispheres (*Figure 4D–F*) and extracted decay time constants ($\tau$, *Figure 4G*). Compatible with our observations, $\tau$ differed significantly across cortical areas ($F_{6,30}$ = 4.49, p=0.006, one-way ANOVA with repeated measures), being larger for frontal than posterior areas. Note that, while it is possible that these coefficients capture autocorrelations introduced by intrinsic GCaMP6f dynamics, there is no reason to believe that this affects our conclusions, as indicator dynamics should be similar across regions. Thus, during the evidence-accumulation task, cortical regions display increasing intrinsic timescales going from visual to frontal areas. This is consistent with previous reports for spontaneous activity and other behavioral tasks (*Chaudhuri et al., 2015*; *Hasson et al., 2008*; *Murray et al., 2014*; *Runyan et al., 2017*) and is compatible with our inactivation findings (*Figures 2 and 3*). Nevertheless, a caveat here is that the auto-regressive coefficients of the encoding model could conceivably be spuriously capturing variance attributable to other behavioral variables not included in the model. For example, our model parameterization implicitly assumes that evidence encoding would be linearly related to the side difference in the number of towers. Although this is a common assumption in evidence-accumulation models (e.g. *Bogacz et al., 2006*; *Brunton et al., 2013*), it might not apply to our case. At face value, however, our findings could suggest that the different intrinsic timescales across the cortex are important for evidence-accumulation computations.

## Discussion

Taken together, our results suggest that distributed cortical areas contribute to sensory-evidence accrual on different timescales. Specifically, brief sub-trial inactivations during performance of a decision-making task requiring seconds-long evidence accumulation resulted in distinct deficits in the weighting of sensory evidence from different points in the stimulus stream. This was such that, on average, the inactivation of frontal cortical areas resulted in larger decreases in the use of evidence occurring further in the past from laser onset compared to posterior regions (*Figures 2 and 3*). Compatible with this, using an encoding model of large-scale cortical dynamics, we found that activity timescales vary systematically across the cortex in a way that mirrors the inactivation results (*Figure 4*).

Our results add to a growing body of literature that has revealed that the cortex of rodents and primates appears to be organized in a hierarchy of temporal processing windows across regions (*Chaudhuri et al., 2015*; *Gao et al., 2020*; *Hasson et al., 2008*; *Ito et al., 2020*; *Murray et al., 2014*; *Runyan et al., 2017*; *Spitmaan et al., 2020*). Specifically, they suggest that the contributions of different cortical areas to decision-making computations are similarly arranged in a temporal hierarchy. A caveat here is that our inactivation findings did not exactly match the same area ordering of integration windows from the widefield imaging neural data, nor were all the inactivation effects monotonic. This could be in part due to technical limitations of the experiments. First, the laser powers we used result in large inactivation spreads, potentially encompassing neighboring regions. Moreover, local inactivation could result in changes in the activity of interconnected regions (*Young et al., 2000*), a possibility that should be evaluated in future studies using simultaneous inactivation and large-scale recordings across the dorsal cortex. At face value, however, the findings could be a reflection of the fact that diverse timescales exist at the level of individual neurons within each region (*Bernacchia et al., 2011*; *Cavanagh et al., 2020*; *Scott et al., 2017*; *Spitmaan et al., 2020*; *Wasmuht et al., 2018*). For example, inactivating an area with multimodal distributions of intrinsic timescales across its neurons could conceivably result in non-monotonic effects of inactivation. In any case, our results point to accrual timescale hierarchies being a significant factor contributing to the widespread recruitment of cortical dynamics during evidence-based decisions, as areas get potentially progressively recruited with increasing integration timescale demands. In the future, this hypothesis should be further probed using tasks that explicitly manipulate the timescales of these decision processes.

Our findings also suggest the possibility that the logic of widespread recruitment of cortical regions in complex, time-extended decisions may in part rely on intrinsic temporal integration properties of local cortical circuits, rather than specific evidence-accumulation mechanisms. For instance, it is possible that simple perceptual decisions primarily engage only the relevant sensory areas because they can be made on the fast intrinsic timescales displayed by these regions (*Zatka-Haas et al., 2021*). Along the same lines, it is conceivable that discrepancies in the literature regarding the effects of perturbing different cortical areas during evidence accumulation stem in part from differences in the timescales of the various tasks (*Erlich et al., 2015*; *Fetsch et al., 2018*; *Hanks et al., 2015*; *Katz et al., 2016*; *Pinto et al., 2019*).

An important remaining question is whether evidence from the different time windows is accumulated in parallel or as a feedforward computation going from areas with short to those with long integration time constants. The parallel scheme would be compatible with recent psychophysical findings in humans reporting confidence of their evidence-based decisions (*Ganupuru et al., 2019*). Conversely, a feedforward transformation would be in agreement with human fMRI findings during language processing (*Yeshurun et al., 2017*) and with a previously published model whereby successive (feedforward) convolution operations lead to progressively longer-lasting responses to sensory evidence (*Scott et al., 2017*). Interestingly, the oculomotor integrator of both fish and monkeys appears to be organized as largely feedforward chains of integration leading to systematically increasing time constants (*Joshua and Lisberger, 2015*; *Miri et al., 2011*), perhaps suggesting that this architecture is universal to neural integrators.

Finally, it is also unclear how and where evidence information from different timescales is combined to yield a final decision. Our simultaneous inactivation experiments (*Figure 3*) suggest that dorsal cortical activity is unevenly weighted, potentially by downstream structures. Candidate regions include the medial prefrontal cortex or subcortical structures such as the striatum and the cerebellum, which have been shown to be causally involved in evidence accumulation (*Bolkan et al., 2022*; *Deverett*

*et al., 2019*; *Yartsev et al., 2018*). Other subcortical candidates are midbrain regions shown to have a high incidence of choice signals in a contrast discrimination task (*Steinmetz et al., 2019*).

Much work remains before obtaining a complete circuit understanding of gradually evolving decisions. Our findings highlight the fact that, much like in memory systems (*Jeneson and Squire, 2012*), the timescale of decision processes is an important feature governing their underlying neural mechanisms, a notion which should be incorporated into both experimental and theoretical accounts of decision making.

# Materials and methods

## Key resources table

| Reagent type (species) or resource | Designation | Source or reference | Identifiers | Additional information |
|---|---|---|---|---|
| Mouse line (*Mus musculus*) | (B6.Cg-Tg(Slc32a1-COP4*H134R/EYFP)8Gfng/J) | The Jackson Laboratory | JAX: 014548 | A.k.a VGAT-ChR2-EYFP |
| Mouse line (*Mus musculus*) | IgS6$^{tm93.1(tetO-GCaMP6f)Hze}$ Tg(Camk2a-tTA)1Mmay/J | The Jackson Laboratory | JAX: 024108 | A.k.a. Ai93-D;CaMKIIα-tTA |
| Mouse line (*Mus musculus*) | B6.129S2-Emx1$^{tm1(cre)Krj}$/J | The Jackson Laboratory | JAX: 005628 | A.k.a. Emx1-IRES-Cre |
| Software, algorithm | Matlab 2015b, 2016b, 2017b | Mathworks | https://www.mathworks.com/products/matlab.html | |
| Software, algorithm | ViRMEn | *Aronov and Tank, 2014* | https://pni.princeton.edu/pni-software-tools/virmen | |
| Software, algorithm | NI DAQmx 9.5.1 | National Instruments | https://www.ni.com/en-us/support/downloads/drivers/download.ni-daqmx.html | |
| Software, algorithm | HCImage | Hamamatsu | https://hcimage.com | |
| Software, algorithm | Scanning laser control software | *Pinto et al., 2019*; *Pinto, 2019* | https://github.com/BrainCOGS/laserGalvoControl | |
| Software, algorithm | Widefield data analysis software | *Pinto et al., 2019*; *Pinto, 2020* | https://github.com/BrainCOGS/widefieldImaging | |
| Software, algorithm | Inactivation analysis | This paper, *Pinto, 2022* | https://github.com/BrainCOGS/PintoEtAl2020_sub-trial_inact | |
| Software, algorithm | Python 3.8 | Python | https://www.python.org | |
| Software, algorithm | Numpy 1.18.1 | *van der Walt et al., 2011* | https://www.numpy.org | |
| Software, algorithm | Scipy 1.4.1 | *Virtanen et al., 2020* | https://www.scipy.org | |
| Software, algorithm | Deepdish 0.3.4 | University of Chicago, *Larsson, 2022* | https://github.com/uchicago-cs/deepdish | |
| Software, algorithm | Statsmodels 0.11.0 | *Skipper, 2010* | https://www.statsmodels.org | |
| Software, algorithm | Matplotlib 3.1.3 | *Hunter, 2007* | https://www.matplotlib.org | |
| Software, algorithm | Pandas 1.0.1 | *McKinney, 2010* | https://www.pandas.pydata.org | |
| Software, algorithm | Pingouin 0.3.8 | *Vallat, 2018* | https://pingouin-stats.org | |
| Software, algorithm | Mat7.3 | Simon Kern, *Kern, 2022* | https://github.com/skjerns/mat7.3 | |
| Software, algorithm | Pymer4 | *Jolly, 2018* | | |

## Animals and surgery

All procedures were approved by the Institutional Animal Care and Use Committee at Princeton University (protocols 1910–15 and 1910–18) and were performed in accordance with the Guide for the Care and Use of Laboratory Animals (National Research Council, 2011). We used both male and female VGAT-ChR2-EYFP mice aged 2–16 months (B6.Cg-Tg[Slc32a1-COP4*H134R/EYFP]8Gfng/J,

Jackson Laboratories, stock # 014548, n=28). Part of the inactivation data from some of these animals was collected in the context of previous work (*Pinto et al., 2019*), but the analyses reported here are completely novel. The mice underwent sterile surgery to implant a custom titanium headplate and optically clear their intact skulls, following a procedure described in detail elsewhere (*Pinto et al., 2019*). Briefly, after exposing the skull and removing the periosteum, successive layers of cyanoacrylate glue (krazy glue, Elmers, Columbus, OH) and diluted clear metabond (Parkell, Brentwood, NY) were applied evenly to the dorsal surface of the skull and polished after curing using a dental polishing kit (Pearson dental, Sylmar, CA). The headplate was attached to the cleared skull using metabond, and a layer of transparent nail polish (Electron Microscopy Sciences, Hatfield, PA) was applied and allowed to cure for 10–15 min. The procedure was done under isoflurane anesthesia (2.5% for induction, 1.5% for maintenance). The animals received two doses of meloxicam for analgesia (1 mg/kg I.P or S.C.), given at the time of surgery and 24 hr later, as well as peri-operative I.P. injections of body-temperature saline to maintain hydration. Body temperature was maintained constant using a homeothermic control system (Harvard Apparatus, Holliston, MA). The mice were allowed to recover for at least 5 days before starting behavioral training. After recovery they were restricted to 1–2 mL of water per day and extensively handled for another 5 days, or until they no longer showed signs of stress. We started behavioral training after their weights were stable and they accepted handling. During training, the full allotted fluid volume was typically delivered within the behavioral session, but supplemented if necessary. The mice were weighed and monitored daily for signs of dehydration. If these were present or their body mass fell below 80% of the initial value, they received supplemental water until recovering. They were group housed throughout the experiment and had daily access to an enriched environment (*Pinto et al., 2018*). The animals were trained 5–7 days/week.

The analysis reported in *Figure 4* (widefield Ca$^{2+}$ imaging) is from data collected in the context of a previous study (*Pinto et al., 2019*), although the analysis is novel. The data was from six male and female mice from triple transgenic crosses expressing GCaMP6f under the CaMKIIα promoter from the following two lines: Ai93-D; CaMKIIα-tTA [IgS6$^{tm93.1(tetO-GCaMP6f)Hze}$ Tg[Camk2a-tTA]1Mmay/J, Jackson Laboratories, stock # 024108] and Emx1-IRES-Cre [B6.129S2-Emx1$^{tm1(cre)Krj}$/J, Jackson Laboratories, stock # 005628]. These animals also underwent the surgical procedure described above.

## Virtual reality apparatus

The mice were trained in a VR environment (*Figure 1A*) described in detail elsewhere (*Pinto et al., 2018*). Briefly, they sat on an 8-inch hollow Styrofoam ball that was suspended by compressed air at ~60 p.s.i, after passing through a laminar flow nozzle to reduce noise (600.326.5 K.BC, Lechler, St. Charles, IL). They were head-fixed such that their snouts were aligned to the ball equator and at a height such that they could run comfortable without hunching, while still being able to touch the ball with their full paw pads (corresponding to a headplate-to-ball height of ~1 inch for a 25 g animal). Ball movements were measured using optical flow sensors (ADNS-3080 APM2.6) and transformed into virtual world displacements using custom code running on Arduino Due (https://github.com/sakoay/AccumTowersTools/tree/master/OpticalSensorPackage, *Koay, 2018*). The ball sat on a custom 3D-printed cup that contained both the air outlet and the movement sensor. The VR environment was projected onto a custom-built toroidal Styrofoam screen using a DLP projector (Optoma HD141X, Fremont, CA) at a refresh rate of 120 Hz and a pixel resolution of 1024 × 768. The screen spanned ~270° of azimuth and ~80° of elevation in the mouse's visual field. The whole setup was enclosed in a custom-built sound-attenuating chamber. The VR environment was programmed and controlled using ViRMEn (*Aronov and Tank, 2014*) (https://pni.princeton.edu/pni-software-tools/virmen), running on Matlab (Mathworks, Natick, MA) on a PC.

## Behavioral task

We trained the mice in the accumulating towers task (*Pinto et al., 2018*). The mice ran down a virtual T-maze that was 3.3 m in length (y), 5 cm in height, and a nominal 10 cm in width (x, though they were restricted to the central 1 cm). The length of the maze consisted of a 30 cm start region to which they were teleported at the start of each trial, followed by a 200 cm cue region and a 100 cm delay region. The cue and the delay region had the same wallpaper designed to provide optical flow. During the cue region, the mice encountered tall white objects (2 × 6 cm, width × height), or towers, that appeared at random locations in each trial at a Poisson rate of 7.7 m$^{-1}$ and 2.3 m$^{-1}$ on the rewarded

and non-rewarded sides, respectively (or 8.0 and 1.6 m$^{-1}$ in some sessions), with a 12 cm refractory period and an overall density of 5 m$^{-1}$. The towers appeared when the mice were 10 cm away from their drawn locations and disappeared 200 ms later (roughly corresponding to the time over which the tower sweeps across the visual field given average running speeds). After the maze stem, the mice turned into one of the two arms (10.5 × 11 × 5 cm, length × width × height), and received a reward if they turned to the arm corresponding to the highest tower count (4–8 μL of 10% v/v sweet condensed milk). This was followed by a 3 s inter-trial interval, consisting of 1 s of a frozen frame of the VR environment and 2 s of a black screen. An erroneous turn resulted in a loud sound and a 12 s timeout.

Each daily behavioral session (~1 hr, ~200–250 trials) started with warm-up trials of a visually guided task in the same maze, in which towers appeared only on the rewarded side and additionally a 30 cm tall visual guide visible from the start of the trial was placed in the arm corresponding to the reward location. The animals progressed to the main task when they achieved at least 85% correct trials over a running window of 10 trials in the warm-up task. During the accumulating-towers task, performance was evaluated over a 40-trial running window, both to assess side biases and correct them using an algorithm described elsewhere (*Pinto et al., 2018*) and to trigger a transition into a 10-trial block of easy trials if performance fell below 55% correct. These blocks consisted of towers only on the rewarded side and were introduced to increase motivation but were not included in the analyses. No optogenetic inactivation was performed during either warm-up or easy-block trials. In the widefield imaging experiments, the behavioral sessions contained several visually guided (warm-up) blocks (*Pinto et al., 2019*). These were excluded from the present analyses.

## Laser-scanning optogenetic inactivation

We used a scanning laser setup described in detail elsewhere (*Pinto et al., 2019*). Briefly, a 473 nm laser beam (OBIS, Coherent, Santa Clara, CA) was directed to 2D galvanometers using a 125-μm single-mode optic fiber optic (Thorlabs, Newton, NJ) and reached the cortical surface after passing through an f-theta scanning lens (LINOS, Waltham, MA). We used a 40 Hz square wave with an 80% duty cycle and a power of 6 mW measured at the level of the skull. This corresponds to an inactivation spread of ~2 mm (*Pinto et al., 2019*). While this may introduce confounds regarding ascribing exact functions to specific cortical areas, we have previously shown that the effects of whole-trial inactivations at much lower powers (corresponding to smaller spatial spreads) are consistent with those obtained at 6 mW. To minimize post-inactivation rebounds, the last 100 ms of the laser pulse consisted of a linear ramp-down of power (*Guo et al., 2014*; *Pinto et al., 2019*). We performed inactivations during the following trial epochs: first, second , or third quarter of the cue region (0–50 cm, 50–100 cm, or 100–150 cm, respectively), first or second half of the cue region (0–100 cm or 100–200 cm, respectively), or delay region (200–300 cm). Thus, the epochs were defined according to the animals' y position in the maze. Because of this, the onset time of the power ramp-down was calculated in each trial based on the current speed and the expected time at which the mouse would reach the laser offset location. The system was controlled using custom-written code in Matlab running on a PC, which sent command analog voltages to the laser and galvanometers through NI DAQ cards. This PC received instructions for laser onset, offset, and galvanometer position from the ViRMEn PC through digital lines.

We targeted a total of nine area combinations, either consisting of homotopic bilateral pairs or multiple bilateral locations. The galvanometers alternated between locations at 200 Hz (20 mm travel time: ~250 μs) and in the case of more than two locations, the sequence of visited locations was chosen to minimize travel distance. The inactivated locations were defined based on stereotaxic coordinates using bregma as reference, as follows:

- Primary visual cortex (V1): –3.5 AP, 3 ML
- Medial secondary visual cortex (mV2, ~area AM): –2.5 AP, 2.5 ML
- Posterior parietal cortex (PPC): –2 AP, 1.75 ML
- Retrosplenial cortex (RSC): –2.5 AP, 0.5 ML
- Posteromedial portion of the premotor cortex (mM2): 0.0 AP, 0.5 ML
- Anterior portion of the premotor cortex (aM2): +3 AP, 1 ML
- Primary motor cortex (M1): +1 AP, 2 ML
- Posterior cortex: V1, mV2, PPC, and RSC
- Frontal cortex: mM2, aM2, and M1

To ensure consistency in bregma location across behavioral sessions, the experimenter set bregma on a reference image and for each session the current image of the mouse's skull was registered to this reference using rigid transformations. Different sessions contained different combinations of areas and inactivation epochs, resulting in partially overlapping mice and sessions for each condition. The probability of inactivation trials, therefore, varied across sessions, ranging from a total of 0.15–0.35 across conditions, and from 0.02 to 0.15 per condition. In our experience, capping the probability at ~0.35 is important to maintain motivation throughout the behavioral session.

## Widefield Ca²⁺ imaging

Details on the experimental setup and data preprocessing can be found elsewhere (*Pinto et al., 2019*). Briefly, we used a tandem-lens macroscope (1×–0.63× planapo, Leica M series, Wetzlar, Germany) with alternating 410 nm and 470 nm LED epifluorescence illumination for isosbestic hemodynamic correction and collected 525 nm emission at 20 Hz, using an sCMOS (OrcaFlash4.0, Hamamatsu, Hamamatsu City, Japan), with an image size of 512 × 512 pixels (pixel size of ~17 µm). Images were acquired with HCImage (Hamamatsu) running on a PC and synchronized to the behavior using a data acquisition-triggering TTL pulse from another PC running ViRMEn, which in turn received analog frame exposure voltage traces acquired through a DAQ card (National Instruments, Austin, TX) and saved in the behavioral log file. The image stacks were motion-corrected by applying the x-y shift that maximized the correlation between successive frames, and then were spatially binned to a 128 × 128 pixel image (~68 × 68 µm). The fluorescence values from pixels belonging to different anatomical ROIs were averaged into a single trace, separately for 410 nm ($F_v$) and 470 nm excitation ($F_b$). After applying a heuristic correction to $F_v$ (*Pinto et al., 2019*), we calculated fractional fluorescence changes as $R=F/F_0$, where $F_0$ for each excitation wavelength was calculated as the mode of all $F$ values over a 30 s sliding window with single-frame steps. The final $\Delta F/F$ was calculated using a divisive correction, $\Delta F/F=R_b/R_v − 1$. ROIs were defined based on the Allen Brain Mouse Atlas (ccv3). We first performed retinotopic mapping to define visual areas and used the obtained maps to find, for each mouse, the optimal affine transformation to the Allen framework.

## Data analysis

All analyses of the behavioral effects of cortical inactivations were performed in Python 3.8. Linear encoding model fitting of widefield data was performed in Matlab, and the results were analyzed in Python.

## Behavioral data selection

Because of the warm-up and easy-block trials, the sessions are naturally organized into a block structure, such that the duration of each block of the accumulating-towers task is of at least 40 trials (see above). We selected all trials from blocks in which the control (laser off) performance was at least 60% correct, collapsed over all levels of sensory evidence. After block selection, we excluded trials in which the animals failed to reach the end of the maze, or in which the total traveled distance exceeded the nominal maze length by more than 10% (*Pinto et al., 2018*; *Pinto et al., 2019*). These selection criteria yielded a total of 929 optogenetic inactivation sessions from 28 mice (average ~33 /mouse), corresponding to 108,940 control (laser off) trials, and 29,825 inactivation trials (average ~552/condition, see *Figure 1—source data 1*). Twenty-five sessions from six mice were selected for widefield imaging data analysis.

## Analysis of behavioral data

### Overall performance

We calculated overall performance as the percentage of trials in which the mice turned to the side with the highest tower counts, separately for control and inactivation trials.

### Running speed

Speed was calculated for each inactivation segment using the total x-y displacement. We compared laser-induced changes in speed to control trials from the same maze segment.

## Psychometric curves

We computed psychometric curves separately for control and inactivation trials by plotting the percentage of right-choice trials as a function of the difference in the number of right and left towers (#R – #L, or Δ). Δ was binned in increments of five between –15 and 15, and its value defined as the average Δ weighted by the number of trials. We fitted the psychometric curves using a four-parameter sigmoid:

$$p_R = b + \frac{a}{1+\exp(-(\Delta - \Delta_0)/\lambda)}$$

## Evidence-weighting curves in space

To assess how mice weighted sensory evidence from different segments of the cue region, we performed a logistic regression analysis in which the probability of a right choice was predicted from a logistic function of the weighted sum of the net amount of sensory evidence from each of four equally spaced segments (10–200 cm, since no towers can occur before y=10):

$$p_R = \frac{1}{1+\exp(-(\beta_0 + \sum_{i=1}^{4} \beta_i \Delta_i))}$$

where Δ = # right – # left towers calculated separately for each segment. These weighting functions were calculated separately for 'laser on' and 'laser off' trials. To quantify the laser-induced changes in evidence weighting, we simply subtracted the 'laser on' from the 'laser off' curves, such that negative values indicate smaller evidence weights in the 'laser on' condition. Bin sizes were chosen to match the resolution of our inactivation epochs.

## Evidence-weighting curves in time

To directly quantify how our manipulations impacted the weighting of sensory evidence in time relative to inactivation onset and offset, while accounting for variability introduced by different mice and epochs in the trial, we fitted a mixed-effects logistic regression model to the mice's choices using the Pymer4 package for Python (*Jolly, 2018*):

$$p_R = \frac{1}{1+\exp(-(\beta_0^C + E^C + \beta_0^L + E^L + R))} \tag{1}$$

where $p_R$ is the probability of making a right side choice, $\beta_0^C$ and $\beta_0^L$ are fixed-effects bias terms for control and laser trials, respectively, $R$ denotes random effects (see below), and $E^X$ is the sensory-evidence fixed-effects terms for control (X=C) and laser-on (X=L) trials:

$$E^X = \sum_{i=1}^{3} \beta_i^{Pre} \Delta_i^{Pre} + \beta^{During} \Delta^{During} + \sum_{i=1}^{2} \beta_i^{Post} \Delta_i^{Post}$$

where $\beta_i^Y$ are the evidence weights and $\Delta_i^Y$ = # right – # left towers calculated separately for each 0.5 s time bin $i$, aligned by the laser onset or offset depending on superscript $Y$. $Y=Pre$ captures evidence occurring before inactivation; thus, evidence time is aligned to laser onset and binned at increasingly negative time values. $Y=During$ captures evidence occurring while the laser is on; thus, evidence time is aligned to laser offset and binned at negative time values. Finally, $Y=Post$ captures evidence occurring after the laser is off; thus, evidence time is aligned to laser offset and binned at positive time values. For control trials, we used dummy laser onset times defined as the time in the control trial corresponding to when the animal crossed the y position value where the laser was on in the nearest inactivation trial. $\Delta_i^Y$ values were z-scored. The normalized coefficients in *Figures 2 and 3*, and *Figure 2—figure supplement 2* were calculated as,

$$(E^L - E^C)/E^C$$

We included two classes of random effects, mouse identity, and inactivation epoch. (Here, 'epoch' is not defined relative to the laser timing but is relative to the start of the maze, as in the schematic of *Figure 1D*.) Mouse identity had both a single random intercept (i.e. side bias) and a random slope for each model coefficient, whereas epoch only had a random intercept corresponding to side bias.

$$R = \sum_{e=1}^{6} \beta_{0,e} + \sum_{m=1}^{n} \beta_{0,m}^{C} + \sum_{m=1}^{n} \beta_{0,m}^{L} + \sum_{m=1}^{n} E_{m}^{C} + \sum_{m=1}^{n} E_{m}^{L} \quad (3)$$

where $e$ is the index over inactivation epochs for a given area, $m$ is the index over mice, and $n$ is the number of mice for a given area. $\beta_{0,i}$ is the bias (random intercept) terms for the $i$th epoch, $\beta_{0,i}^{x}$ are the bias (random intercept) terms for the $i$th mouse in either control (*C*) or laser (*L*) trials, and $E_{m}^{x}$ are the per-mouse evidence random effects (slopes), with each term as described above in equation (2). $R$ is modeled as a multivariate Gaussian distribution with mean = 0. In other words, the per-mouse random effects are modeled as a noise sample from this zero-mean gaussian, with the fixed effect of equations (1) and (2) capturing tendencies that are common across the mice. The model is fitted using maximum likelihood estimation as described in detail elsewhere (*Bates et al., 2015*).

All models were fitted using 10-fold cross-validation. We analyzed the best of the 10 models for each condition, defined as the one for which we obtained the highest choice prediction accuracy in the 10% of trials not used to fit the data. For the models in *Figure 2*, we also computed coefficients for shuffled data, where we randomized the laser-on labels 30 times while keeping the mouse and condition labels constant, such that we maintained the underlying statistics for these sources of variability. This allowed us to estimate the empirical null distributions for the laser-induced changes in evidence weighting terms.

## Statistics of inactivation effects

Error estimates and statistics for general performance, running speed, and logistic regression weights in space were generated by bootstrapping this procedure 10,000 times, where in each iteration we sampled trials with replacement. p-values for regression weights and general performance were calculated as the fraction of bootstrapping iterations in which the control-subtracted inactivation value was above zero. In other words, we performed a one-sided test of the hypothesis that inactivation decreases performance and evidence weights on decision. For speed, we performed a two-sided test by computing the proportion of iterations where the sign of the laser-induced change in speed differed from that of the empirical data average. The significance of the coefficients in the mixed-effects model of evidence in time was calculated using a t-test based on the coefficient estimate and its standard error. Additionally, for the models in *Figure 2*, we only considered coefficients to be significant if their standard error did not overlap the ±1 SD intervals from the coefficients extracted from the shuffled models. To compare two coefficients from different models (*Figure 3—figure supplement 1*), we used a z-test, calculating the z statistic as follows *Clogg et al., 1995*:

$$z = |(\beta_1 - \beta_2)/\sqrt{(\beta_1 \times SEM_{\beta_1})^2 + (\beta_2 \times SEM_{\beta_2})^2}|$$

To estimate statistical power, we performed a bootstrapping-based power analysis based on the one described by *Guo et al., 2014*. We randomly subsampled the full dataset containing all inactivation conditions. In each subsample, we selected different numbers of inactivation trials regardless of area-epoch combination (50<n<1000, in steps of 25) and added a random subset of control trials such that the relative proportion of control to laser trials was preserved in the subsampled dataset. We then ran the bootstrapping procedure described above to compute laser-induced changes in overall performance combined across all inactivation conditions, extracting p-values for each of the values of n subsamples. We repeated this procedure 10 times. Power was defined as the minimum number of trials required to observe p<0.05 at the empirical effect size pooled across conditions, as defined by the first n where the 2× SEM across the 10 repeats is below 0.05. We obtained an aggregate power of n=250.

## Linear encoding model of widefield data

We fitted Ca²⁺ activity averaged over each anatomically defined ROI with a linear model (*Pinto et al., 2019*; *Pinto and Dan, 2015*; *Scott et al., 2017*). For each trial and y position in the maze, we

extracted ΔF/F (with native 10 Hz sampling frequency) limited to 0≤y≤ 300 cm (i.e. trial start, outcome, and inter-trial periods were not included). Activity was then z-scored across all trials. ΔF/F of each area was modeled as a linear combination of different predictors at different time lags. In addition to the previously used task-event predictors (*Pinto et al., 2019*), we added coupling terms, i.e., the zero-lag activity of the other simultaneously imaged ROIs (*Pillow et al., 2008*; *Runyan et al., 2017*), as well as auto-regressive terms to capture activity auto-correlations that were independent of task events (*Spitmaan et al., 2020*). Finally, we added a term to penalize the L2 norm of the coefficients, i.e., we performed ridge regression. The full model was thus defined as:

$$\Delta F/F(t) = \beta_0 + A + C + T + \lambda \|\vec{B}\|$$

where $\beta_0$ is an offset term, $\lambda$ is the penalty term, and $\|\vec{B}\|$ is the L2 norm of the weight vector. Additionally, A, C, and T are the auto-regressive, coupling, and task terms, respectively:

$$A = \sum_{i=0.1}^{2} \beta_i^{autoregr} \Delta F/F(t-i)$$

$$C = \sum_{j=1}^{15} \beta_j^{coupling} \Delta F/F_j(t)$$

$$T = \sum_{i=0}^{2} \beta_i^{tR} E_{t-i}^{tR} + \sum_{i=0}^{2} \beta_i^{tL} E_{t-i}^{tL} + \sum_{i=0}^{2} \beta_i^{\Delta} E_{t-i}^{\Delta} + \sum_{i=0.3}^{0.3} \beta_i^{\theta} E_{t-i}^{\theta} + \sum_{i=0.3}^{0.3} \beta_i^{d\theta/dt} E_{t-i}^{d\theta/dt} + \sum_{i=0.3}^{0.3} \beta_i^{sp} E_{t-i}^{sp} + \beta^y y + \beta^{ch} ch + \beta^{pch} pch + \beta^{prw} prw$$

In the above equations, $\beta_i^x$ is the encoding weight for predictor x at time lag i (in steps of 0.1 s), where x is either a task event or the activity of the ROI at a previous time point, and $\beta_j^{coupling}$ is the weight for the zero-lag activity for simultaneously imaged ROI j (we had a total of 16 ROIs across the 2 hemispheres). In the task term, $E_{t-i}^j$ is a delta function indicating the occurrence of event x at time t-i. Specifically, tR indicates the occurrence of a right tower, tL of a left tower, Δ=cumulative #R – #L towers, θ is the view angle, dθ/dt is the virtual view angle velocity, sp is the running speed, y is the spatial position in the maze stem (no lags), and ch, pch, and prw are constant offsets for a given trial, indicating upcoming choice, previous choice (+1 for right and –1 for left) and previous reward (1 for reward and –1 otherwise), respectively.

## Cross-validation

The model was fitted using threefold cross-validation. For each of 20 values of the penalty term $\lambda$, we trained the model using two-thirds of the trials (both correct and wrong choices) and tested it on the remaining one-third of trials. We picked the value of $\lambda$ that maximized accuracy and used median accuracy and weight values across all 10 × 3 runs for that $\lambda$. Model accuracy was defined as the linear correlation coefficient between actual ΔF/F and that predicted by the model in the test set.

## Model comparison

We tested three versions of the encoding model, one with just the task term T, another one adding the auto-regressive term A, and the other with the coupling term C in addition to A and T. All versions were fitted using exactly the same cross-validation data partitioning to allow for direct comparison. We averaged cross-validated predictions over hemispheres and sessions for each mouse, performing the comparison with mouse-level data. Statistical significance of the differences between the accuracy of different models was computed using a two-way ANOVA with repeated measures with factors ROI and model type, and individual model comparisons were made using Tukey's post hoc test. Coefficient analysis in *Figure 4* is from the full model, which had the highest performance.

## Quantification of timescales from the model coefficients

To quantify the timescales from the fitted auto-regressive coefficients, for each behavioral session we fitted an exponential decay function to the coefficients between 0.1 and 2 s in the past, normalized to the coefficient at 0.1 s (first bin):

$$B + A\exp(-x/\tau)$$

where B is the offset term, A controls the amplitude of the curve, x is the vector of normalized coefficients, and $\tau$ is the decay time constant. Fits were performed using the non-linear least squares

algorithm. The extracted time constants ($\tau$) were first averaged over hemispheres and sessions for each mouse, and statistics were performed on mouse averages. The significance of the differences in the time constants across regions was assessed by performing a one-way ANOVA with repeated measures, with cortical regions as the factor.

## False discovery rate correction

We corrected for multiple comparisons using a previously described method for false discovery rate correction (**Benjamini and Hochberg, 1995**; **Guo et al., 2014**; **Pinto et al., 2019**). Briefly, p-values were ranked in ascending order, and the $i$th ranked p-value, $P_i$, was deemed significant if it satisfied $P_i \leq (\alpha i)/n$, where $n$ is the number of comparisons and $\alpha$ is the significance level. In our case, $\alpha=0.050$ and 0.025 for one-sided and two-sided tests, respectively.

## Data and code availability

Data analysis code and source code for figures are available at https://github.com/Brain-COGS/PintoEtAl2020_sub-trial_inact (copy archived at swh:1:rev:de1261fff8f39a8aa14c-de34da032384fe3b9144, **Pinto, 2022**). Behavioral data from inactivation experiments is publicly available on figshare.com, doi: 10.6084 /m9.figshare.19543948.

## Acknowledgements

We thank Sue Ann Koay and Kanaka Rajan for discussions, Abigail Russo and E Mika Diamanti for comments on the manuscript, and Samantha Stein and Scott Baptista for technical assistance.

## Additional information

### Funding

| Funder | Grant reference number | Author |
|---|---|---|
| National Institutes of Health | U01NS090541 | Lucas Pinto<br>David W Tank<br>Carlos D Brody |
| National Institutes of Health | U19NS104648 | Lucas Pinto<br>David W Tank<br>Carlos D Brody |
| National Institutes of Health | F32NS101871 | Lucas Pinto |
| National Institutes of Health | K99MH120047 | Lucas Pinto |
| National Institutes of Health | R00MH120047 | Lucas Pinto |
| Simons Foundation | 872599SPI | Lucas Pinto |

The funders had no role in study design, data collection and interpretation, or the decision to submit the work for publication.

### Author contributions

Lucas Pinto, Conceptualization, Data curation, Formal analysis, Funding acquisition, Investigation, Methodology, Project administration, Software, Validation, Visualization, Writing – original draft, Writing – review and editing; David W Tank, Carlos D Brody, Conceptualization, Funding acquisition, Project administration, Supervision, Writing – review and editing

### Author ORCIDs

Lucas Pinto http://orcid.org/0000-0002-0471-9317
David W Tank http://orcid.org/0000-0002-9423-4267
Carlos D Brody http://orcid.org/0000-0002-4201-561X

## Ethics

All procedures were approved by the Institutional Animal Care and Use Committee at Princeton University (protocols 1910-15 and 1910-18) and were performed in accordance with the Guide for the Care and Use of Laboratory Animals. Surgical procedures were done under isoflurane anesthesia. The animals received two doses of meloxicam for analgesia , given at the time of surgery and 24 h later, as well as peri-operative I.P. injections of body-temperature saline to maintain hydration. Body temperature was maintained constant using a homeothermic control system. The mice were allowed to recover for at least 5 days before starting behavioral training. After recovery they were restricted to 1 - 2 mL of water per day and extensively handled for another 5 days, or until they no longer showed signs of stress. We started behavioral training after their weights were stable and they accepted handling. During training, the full allotted fluid volume was typically delivered within the behavioral session, but supplemented if necessary. The mice were weighed and monitored daily for signs of dehydration. If these were present or their body mass fell below 80% of the initial value, they received supplemental water until recovering. They were group housed throughout the experiment, and had daily access to an enriched environment. The animals were trained 5 - 7 days/week.

## Decision letter and Author response

Decision letter https://doi.org/10.7554/eLife.70263.sa1
Author response https://doi.org/10.7554/eLife.70263.sa2

# Additional files

## Supplementary files

• Transparent reporting form

## Data availability

Data analysis code and source code for figures is available at https://github.com/Brain-COGS/PintoEtAl2020_subtrial_inact (copy archived at swh:1:rev:de1261fff8f39a8aa14c-de34da032384fe3b9144). - Each figure contains associated source data containing the numerical data used to generate them - Behavioral data from all inactivation experiments is publicly available on figshare.com, https://doi.org/10.6084/m9.figshare.19543948.

The following dataset was generated:

| Author(s) | Year | Dataset title | Dataset URL | Database and Identifier |
|---|---|---|---|---|
| Pinto L, Tank DW, Brody CD | 2022 | Behavioral data in the accumulating-towers task with optogenetic inactivation of 9 sets of cortical regions during six different trial epochs | https://doi.org/10.6084/m9.figshare.19543948 | figshare, 10.6084/m9.figshare.19543948 |

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
