## [Editor Report]

Pinto and colleagues used brief optogenetic silencing to study the contributions of different cortical areas in an evidence accumulation task in mice. The authors show that silencing of frontal regions affected evidence accumulation on a longer timescale than that of posterior regions, providing evidence indicating the relation between cortical functions and intrinsic timescales.

---

## [Decision Letter]

**Decision letter after peer review:**

Thank you for submitting your article "Multiple timescales of sensory-evidence accumulation across the dorsal cortex" for consideration by *eLife*. Your article has been reviewed by 3 peer reviewers, one of whom is a member of our Board of Reviewing Editors, and the evaluation has been overseen by Michael Frank as the Senior Editor. The following individual involved in review of your submission has agreed to reveal their identity: Gidon Felsen (Reviewer #3).

Essential revisions:

Previous studies have indicated that neurons in different cortical areas have different intrinsic timescales. In this study, Pinto and colleagues aimed at establishing the functional significance of intrinsic timescales across cortical regions by performing optogenetic silencing of cortical areas in an evidence accumulation task in mice. The authors observed that optogenetic silencing reduced the weight of sensory evidence primarily during silencing, but also preceding time windows in some cases, suggesting that inactivation of frontal cortical regions had long-lasting effects than that of posterior cortical regions. This study provides important results addressing the relation between cortical functions and intrinsic timescales.

The reviewers agreed that this study addresses an important question, and the authors performed sophisticated experiments, and collected a large amount of data. The results are presented clearly and the manuscript is well-written. All the reviewers thought that the results are potentially of great interest to a wide audience. However, the reviewers found several substantive issues which reduced the confidence on the authors' conclusions. These issues need to be addressed before publication of this study at *eLife*.

In particular, the following points have been identified as essential issues:

1. The presented analysis does not consider a large variability that exists at the level of individual animals. There is also some variability across conditions (e.g. photoinhibition of different epochs). Furthermore, the statistical analyses presented in the manuscript often rely on a small number of samples, and the sample size is not equal across the conditions (n = 6, 4, 3 for y = 0, 50, 100, respectively). Because of these issues, we felt that the main conclusion needs to be supported by further analysis investigating these variabilities, and careful discussions of these potential caveats.

2. The authors claim that the optogenetic silencing primarily affected the evidence-accumulation computation, but not other decision-related processes. The reviewers found this claim to be not strongly supported by the data. From the presented data, whether silencing specifically affected the evidence-accumulation process, not just passing the evidence to an accumulation process, remains unclear. Furthermore, silencing affects running speed (thus, indicates effects other than accumulation process). Also, the reviewers thought that alternative possibilities have not been fully examined.

3. Optogenetic silencing sometimes increased the running speed. This can potentially reduce the time spent in each location, and may affect the acquisition sensory information. It is important that the reduced regression weight is not the side effect of reduce time spent in each location. Furthermore, some analysis based on time, not just locations, would be very helpful.

More detailed comments and suggestions on the above issues are included in the individual reviewers' comments.

*Reviewer #2 (Recommendations for the authors):*

1) Overall, the inactivation effect is highly variable across brain regions and conditions. For example, in Figure 1-Supp 2, silencing mV2 and RSC during the 3rd quarter of the cue region reduce weighting 100 cm back, but the effect is not replicated when silencing is extended in time (2nd half of the cue region). The effect is yet different when silencing the posterior cortical regions, which covers mV2 and RSC. There are many cases like this. What is this variability due to? Is this degree of variability expected from behavioral variability? It is difficult to evaluate how robust the behavioral deficits are without an estimate of the expected variability and false positive rate.

2) The conclusion that inactivation primarily affects evidence accumulation is based on weights from the logistic regression. A drop in weights of the sensory evidence presumably means the stimulus information is lost. However, there could be other reasons weights could drop. For example, if mice stop engage in the task after photostimulation, this could presumably lower the weights since mice no longer base their choice on the sensory stimulus. The analysis of weights after photostimulation provides a nice control (Figure 2-Supp2). However, several areas do show prospective deficits in weighting of future evidence, although this is not observed in all areas. Prospective deficits could be consistent with mice stop performing the task. This possibility should be ruled out.

3) Some additional analyses could further corroborate the interpretation that the deficit is specifically in evidence accumulation. For example, if the inactivation selectively abolishes the memory of prior evidence, stimuli presented thereafter should still be integrated and a model based the evidence after the photostimulus should predict choice. If so, this could strengthen the interpretation that the deficits are specific to the accumulated evidence. Otherwise, it could suggest inactivation is degrading performance for other reasons.

4) In general, I could not find information on how well the logistic regression predicts choice.

5) The main result of the paper (Figure 2) is based on effects averaged across different inactivation conditions (different epochs). However, I wonder if it makes sense to combine conditions like this. One, I wonder if this could hide areas that are involved during specific epochs of the task. The text states that "…aligned curves from different epochs were fairly consistent (Figure 2B)", but it is not clear how this is quantified and compared to what reference. Two, I wonder if this pooling would violate assumptions of statistical tests given data now comes from distinct sources, rather than being repeated observations.

6) The analysis of calcium dynamics are based on the autoregressive component of the GLM model. This is counterintuitive because that component is not related to the stimulus or the task. If the claim is that evidence accumulation is related to the timescale of neural dynamics, shouldn't the analysis focus on the coefficients for E_δ (cumulative #R – #L towers), i.e. the component of the dynamics that encodes the stimulus?

7) In a couple of places in the text, I feel the claims should be weakened as they go beyond the data. For example,

a. Intro: "… provide the first casual demonstration that this hierarchy [of timescale] is important for cognitive behavior." A similar statement is in the 2nd paragraph of discussion. I suggest changing the framing. The experiments do not manipulate the timescale of cortical regions. The relationship with the observed behavioral deficit is correlative.

b. Page 11, "This suggests that signals from the different dorsal cortical areas could be combined by downstream regions in a near-linear fashion. Candidate regions include … " The following paragraph is perhaps more suitable for discussion since the experiments do not probe subcortical regions. Also see comment 8 below. The effects of combined-area inactivation in fact appear to be qualitatively different from the average of single area silencing.

c. Page 13, "…the different intrinsic timescales across the cortex support evidence integration over time windows of different durations." For the same reason as in comment (a) above, I suggest rephrasing or removing this framing.

d. Abstract and intro, "inactivation of different areas primarily affected the evidence-accumulation per se, rather than other decision-related process". It seems the results do not examine other decision-related process besides the weighting of sensory evidence.

e. The text claims the spatial resolution of inactivation is 1.5-2mm. This is somewhat misleading. In Figure S2 of Pinto 2019, 60% of neurons are silenced at this light intensity at 2mm from light center. This broad inactivation is also consistent with the characterization from the Svoboda lab (Li et al., *eLife* 2019), which suggests that the spread of inactivation at 6 mW extends well beyond 2 mm in radius.

8) In Figure 2-Supp 3, the effects of posterior vs frontal cortex inactivation do not appear to be very different from each other. This is somewhat different from the averages of single area effects. In general, the statistical tests in the paper do not directly compare the effects of posterior cortex inactivation vs. frontal cortex inactivation. A more appropriate test for the key conclusion should be an interaction of y-position dependence with cortical regions.

9) The explanation of power analysis is not very clear (page 26-27). How are the control trials subsampled at different number of inactivation trials? What does it mean to bootstrap all the inactivation conditions together? At what effect size is n=250 sufficient to detect the effect?

10) The non-monotonic effect of cluster 3 (V1 and RSC) in Figure 2c is counterintuitive. The effect seems to be present in several individual conditions in Figure 1-Supp 2. However, other conditions don't show this (e.g. delay epoch inactivation). The text states that the effect is potentially compatible with findings that multiple timescales exist in a single region. Please explain this notion more clearly and how it could lead to no deficit for recent stimulus information but deficits for distant stimulus memory.

11) Mice speed up during photostimulation in nearly all conditions (Figure 2-Supp 1). Are mice responding to the light? Ideally, a negative control could be included to show there are no non-specific effects of photostimulation when analyzed in the logistic regression. This could be done by photostimulation in GFP mice or by inactivation a cortical region not involved in the behavior.

*Reviewer #3 (Recommendations for the authors):*

Related to the above comment on aggregating data across mice, the presentation of the data would be more transparent if mouse-by-mouse results were shown, where possible (like they are in Figure 1B,C; Figure 1-table S1 is also helpful). For example, symbols for individual mice could be shown in Figure 1E instead of (or in addition to) the mean across mice. Presumably change in performance was calculated within mice and then averaged, rather than averaging laser on and laser off performance across mice and then taking the difference between the two. But the description ("inactivation-induced change in overall % correct performance for each inactivation epoch, for data combined across mice", line 119) could apply to either analysis.

---

## [Author Response]

Essential revisions:Previous studies have indicated that neurons in different cortical areas have different intrinsic timescales. In this study, Pinto and colleagues aimed at establishing the functional significance of intrinsic timescales across cortical regions by performing optogenetic silencing of cortical areas in an evidence accumulation task in mice. The authors observed that optogenetic silencing reduced the weight of sensory evidence primarily during silencing, but also preceding time windows in some cases, suggesting that inactivation of frontal cortical regions had long-lasting effects than that of posterior cortical regions. This study provides important results addressing the relation between cortical functions and intrinsic timescales.The reviewers agreed that this study addresses an important question, and the authors performed sophisticated experiments, and collected a large amount of data. The results are presented clearly and the manuscript is well-written. All the reviewers thought that the results are potentially of great interest to a wide audience. However, the reviewers found several substantive issues which reduced the confidence on the authors' conclusions. These issues need to be addressed before publication of this study at eLife.

We are glad the reviewers found our work to be of potentially great interest, and we thank them for their valuable, thorough and constructive critique of our manuscript. We believe they raised a number of important issues, which prompted us to significantly revise our analytical approaches and writing. Specifically, we have made the following major changes to the manuscript:

– We have replaced the analysis of inactivation effects on evidence weighting for a mixed-effects logistic regression model that is parameterized in time instead of space. This new parameterization better separates the effects on sensory evidence that occurs before, during or after the inactivation, all within the same models. Moreover, variability from individual mice is now explicitly accounted for as random effects. Further, we also combine different epochs in the same model by adding them as additional random effects. These changes simultaneously address major concerns about inter-mouse variability, low-n statistics on conditions rather than mice, and potential confounds from laser-induced changes and inter-trial variability in running speed. Overall, the results from the new analysis largely match our previous analysis, but the increased statistical power and better parameterization allowed us to clear up some findings.

– Individual mouse data are now displayed where relevant.

– We have added analyses of stimulus-related coefficients in the linear encoding models of widefield data.

– We have removed our claims of causality and exclusive inactivation effects on evidence accumulation, and added more thorough discussions of the caveats around the interpretation of our findings.

We hope the reviewers and editors will agree that we have satisfactorily addressed their concerns, and as a result our manuscript is much improved. We provide detailed responses to the individual points below.

In particular, the following points have been identified as essential issues:1. The presented analysis does not consider a large variability that exists at the level of individual animals. There is also some variability across conditions (e.g. photoinhibition of different epochs). Furthermore, the statistical analyses presented in the manuscript often rely on a small number of samples, and the sample size is not equal across the conditions (n = 6, 4, 3 for y = 0, 50, 100, respectively). Because of these issues, we felt that the main conclusion needs to be supported by further analysis investigating these variabilities, and careful discussions of these potential caveats.

We have entirely replaced this analysis for a mixed-effects logistic regression approach in which the fixed effects are weights of sensory evidence in time, and random effects are mice and conditions. Thus, we now explicitly model the variability introduced by these two factors, which allows us to focus on the effects that are common across mice and conditions. Additionally, we now perform statistical analyses on coefficients using metrics based on their error estimates from the model fitting procedure, such that all estimates come from the same sample size and take into account the full data (t- and z-tests, as explained in more detail in Materials and methods, line 665). Finally, to further estimate variability introduced by mice and conditions (epochs), we have devised a shuffling procedure where laser-on labels are shuffled while maintaining mouse and condition statistics. We use this procedure to estimate the empirical null distributions of inactivation-induced changes in evidence weights, which we use to further assess statistical significance of the effects. These new analyses are presented in Figures 2, 3 and corresponding supplements. We have also added text to be more explicit about these sources of variability (line 169):

“(…) to account for the inter-animal variability we observed, we used a mixed-effects logistic regression approach, with mice as random effects (see Materials and methods for details), thus allowing each mouse to contribute its own source of variability to overall side bias and sensitivity to evidence at each time point, with or without the inactivations. We first fit these models separately to inactivation epochs occurring in the early or late parts of the cue region, or in the delay (y ≤ 100 cm, 100 < y ≤ 200 cm, y > 200 cm, respectively). We again observed a variety of effect patterns, with similar overall laser-induced changes in evidence weighting across epochs for some but not all tested areas (Figure 2—figure supplement 1). Such differences across epochs could reflect dynamic computational contributions of a given area across a behavioral trial. However, an important confound is the fact that we were not able to use the same mice across all experiments due to the large number of conditions (Figure 1–table supplement 1), such that epoch differences (where epoch is defined as time period relative to trial start) could also simply reflect variability across subjects. To address this, for each area we combined all inactivation epochs in the same model, adding them as additional random effects, thus allowing for the possibility that inactivation of each brain region at each epoch would contribute its own source of variability to side bias; different biases from mice perturbed at different epochs would then be absorbed by this random-effects parameter. We then aligned the timing of evidence pulses to laser onset and offset within the same models, as opposed to aligning with respect to trial start. This alignment combined data from mice inactivated at different epochs together, further ameliorating potential confounds from any mouse x epoch-specific differences. Each fixed-effects data point in figures below (Figures 2, 3, solid colors) thus reflects tendencies common across mice, not individual mouse effects; the latter are shown as the random effects (faded colors). This approach allowed us to extract the common underlying patterns of inactivation effects on the use of sensory evidence towards choice, while simultaneously accounting for inter-subject and inter-condition variability.”

2. The authors claim that the optogenetic silencing primarily affected the evidence-accumulation computation, but not other decision-related processes. The reviewers found this claim to be not strongly supported by the data. From the presented data, whether silencing specifically affected the evidence-accumulation process, not just passing the evidence to an accumulation process, remains unclear. Furthermore, silencing affects running speed (thus, indicates effects other than accumulation process). Also, the reviewers thought that alternative possibilities have not been fully examined.

We agree with the reviewers that our previous modeling approach did not allow us to adequately separate between these different processes, both because it confounded sensory processing and sensory memory and because effects on post-laser sensory evidence were part of a separate analysis that did not take into account pre-laser evidence weights. To address these shortcomings, not only did we fit a new model in time rather than space, but we also separated evidence occurring before, during or after inactivation within the same model. We did so by aligning evidence time to either laser onset or offset (see Materials and methods, page 28, line 614, for details). As we now explain in the main text (line 156):

“We reasoned that changes in the weighting of sensory evidence occurring before laser onset would primarily reflect effects on the memory of past evidence, while changes in evidence occurring while the laser was on would reflect disruption of processing and/or very short-term memory of the evidence. Finally, changes in evidence weighting following laser offset would potentially indicate effects on processes beyond accumulation per se, such as commitment to a decision. For example, a perturbation that caused a premature commitment to a decision would lead to towers that appeared subsequent to the perturbation having no weight on the animal’s choice. Although our inactivation epochs were defined in terms of spatial position within the maze, small variations in running speed across trials, along with the moderate increases in running speed during inactivation, could have introduced confounds in the analysis of evidence as a function of maze location (Figure 1—figure supplement 2). Thus, we repeated the analysis of Figure 1C but now with logistic regression models, built to describe inactivation effects for each area, in which net sensory evidence was binned in time instead of space. (…) We then aligned the timing of evidence pulses to laser onset and offset within the same models, as opposed to aligning with respect to trial start.”

We would also like to emphasize that we consider the evidence-accumulation process to involve both “passing evidence to the accumulator” and “accumulating / remembering it.” To make this more explicit, we have added the following sentence to the introduction (line 48):

“[evidence accumulation] involves remembering a running tally of evidence for or against a decision, updating that tally when new evidence becomes available, and making a choice based on the predominant evidence.”

Throughout our description of results, we now more carefully outline whether the findings support a role in sensory-evidence processing, memory, or both, as well as post-accumulation processes manifesting as decreases in the weight of sensory evidence after laser offset. For example, our new analyses have more clearly shown prospective changes in evidence use when M1 and mM2 were silenced, compatible with the latter. We also agree with the reviewers that we cannot completely rule out other untested sources of behavioral deficits beyond the aforementioned decision processes. Thus, we have removed all statements to the effect that only evidence accumulation per se was affected. Importantly, though, we believe the new analyses do support the claims that the inactivation of all tested areas strongly affects the accumulation process, even if not exclusively.

3. Optogenetic silencing sometimes increased the running speed. This can potentially reduce the time spent in each location, and may affect the acquisition sensory information. It is important that the reduced regression weight is not the side effect of reduce time spent in each location. Furthermore, some analysis based on time, not just locations, would be very helpful.

The increases in running speed are small in magnitude, averaging only ~8% from control levels across conditions (we have changed the units in the speed figure from cm/s to % of control to highlight that). However, we do agree with the reviewers that this potentially introduces confounds when analyzing the effects in space rather than time. Thus, along with the changes described above, we have replaced the analysis in time by a single model that parametrizes evidence in time, aligned to either laser onset or offset. These analyses are now presented in Figures 2 and 3, and corresponding supplements, and largely confirm our main findings.

More detailed comments and suggestions on the above issues are included in the individual reviewers' comments.Reviewer #2 (Recommendations for the authors):1) Overall, the inactivation effect is highly variable across brain regions and conditions. For example, in Figure 1-Supp 2, silencing mV2 and RSC during the 3rd quarter of the cue region reduce weighting 100 cm back, but the effect is not replicated when silencing is extended in time (2nd half of the cue region). The effect is yet different when silencing the posterior cortical regions, which covers mV2 and RSC. There are many cases like this. What is this variability due to? Is this degree of variability expected from behavioral variability? It is difficult to evaluate how robust the behavioral deficits are without an estimate of the expected variability and false positive rate.

We have now changed our modeling approach to better account for the variability across mice and conditions present in the data. We now use a single mixed-effects model to fit all conditions and mice for a given area, using evidence weights as fixed effects and mice and conditions as random effects. Because of the large number of different conditions (54), not all mice were exposed to all area-epoch combinations. Thus, it is unfortunately not possible to dissociate these two sources of variability. For example, it could be the case that inter-condition variability genuinely reflects some dynamic process, such that the effect of inactivating a given area is different depending on the inactivation epoch. While this possibility is interesting, we cannot probe it conclusively because inter-condition variability could trivially arise from differences between mice exposed to different conditions. We now discuss this explicitly in the Results section (line 174):

“(…) We again observed a variety of effect patterns, with similar overall laser-induced changes in evidence weighting across epochs for some but not all tested areas (Figure 2—figure supplement 1). Such differences across epochs could reflect dynamic computational contributions of a given area across a behavioral trial. However, an important confound is the fact that we were not able to use the same mice across all experiments due to the large number of conditions (Figure 1–table supplement 1), such that epoch differences (where epoch is defined as time period relative to trial start) could also simply reflect variability across subjects. To address this, for each area we combined all inactivation epochs in the same model, adding them as additional random effects, thus allowing for the possibility that inactivation of each brain region at each epoch would contribute its own source of variability to side bias; different biases from mice perturbed at different epochs would then be absorbed by this random-effects parameter. We then aligned the timing of evidence pulses to laser onset and offset within the same models, as opposed to aligning with respect to trial start. This alignment combined data from mice inactivated at different epochs together, further ameliorating potential confounds from any mouse x epoch-specific differences. (…) Thus, this approach allowed us to extract the common underlying patterns of inactivation effects on the use of sensory evidence towards choice, while simultaneously accounting for inter-subject and inter-condition variability.”

Following the reviewer's excellent suggestion, we have also devised a procedure to estimate the empirical null distribution of laser effects, taking into account the variability in our data. We then used that distribution to further constrain effect significance. This is explained in Materials and methods (lines 652, 667):

“For the models in Figure 2, we also computed coefficients for shuffled data, where we randomized the laser-on labels 30 times while keeping the mouse and condition labels constant, such that we maintained the underlying statistics for these sources of variability. This allowed us to estimate the empirical null distributions for the laser-induced changes in evidence weighting terms. (…) Significance of the coefficients in the mixed-effects model of evidence in time were calculated using a t-test based on the coefficient estimate and its standard error. Additionally, for the models in Figure 2 we only considered coefficients to be significant if their standard error did not overlap the ± 1 SD intervals from the coefficients extracted from the shuffled models.”

Finally, we note that all of our p-values are in fact corrected for a false discovery rate using Bejamini and Hochberg's method (1995), as described in Materials and methods (line 739):

"We corrected for multiple comparisons using a previously described method for false discovery rate (FDR) correction (Benjamini and Hochberg, 1995; Guo et al., 2014; Pinto et al., 2019). Briefly, p-values were ranked in ascending order, and the *i*th ranked p*-*value, *P_i_*, was deemed significant if it satisfied π ≤ (*αi*)/*n*, where *n* is the number of comparisons and *α* is the significance level. In our case, *α* = 0.050 and 0.025 for one-sided and two-sided tests, respectively."

2) The conclusion that inactivation primarily affects evidence accumulation is based on weights from the logistic regression. A drop in weights of the sensory evidence presumably means the stimulus information is lost. However, there could be other reasons weights could drop. For example, if mice stop engage in the task after photostimulation, this could presumably lower the weights since mice no longer base their choice on the sensory stimulus. The analysis of weights after photostimulation provides a nice control (Figure 2-Supp2). However, several areas do show prospective deficits in weighting of future evidence, although this is not observed in all areas. Prospective deficits could be consistent with mice stop performing the task. This possibility should be ruled out.3) Some additional analyses could further corroborate the interpretation that the deficit is specifically in evidence accumulation. For example, if the inactivation selectively abolishes the memory of prior evidence, stimuli presented thereafter should still be integrated and a model based the evidence after the photostimulus should predict choice. If so, this could strengthen the interpretation that the deficits are specific to the accumulated evidence. Otherwise, it could suggest inactivation is degrading performance for other reasons.

We will address points 2 and 3 together. We agree with the reviewer that our previous modeling approach did not allow us to adequately separate between these different processes, both because it confounded sensory processing and sensory memory and because effects on post-laser sensory evidence were part of a separate analysis that did not take into account pre-laser evidence weights. However, we believe that our new modeling approach addresses these shortcomings by separating evidence occurring before, during or after inactivation *within the same model*. Thus, because choice can be predicted based on both pre- and post-laser evidence, we can now compare the effects on these directly (Figures 2, 2-S1, 2-S2, 3). With two exceptions, we did not observe any significant deficits in prospective evidence weighting. As the reviewer points out, this suggests that the mice perform the task normally after the inactivation in these cases. We did observe post-inactivation changes in evidence use for M1 and mM2, although these were milder than the decreases in pre-laser evidence weighting (Figure 2). We believe that this suggests a role for these areas in both accumulation and post-accumulation processes. Of course, this does suggest that evidence accumulation is not the only affected computation. We also agree that we have not completely ruled out other potential sources of behavioral deficits. Thus, we have removed all statements to the effect that only evidence accumulation per se was affected. Importantly, though, we believe the new analyses do support the claims that the inactivation of all tested areas strongly affects the accumulation process, even if not exclusively.

4) In general, I could not find information on how well the logistic regression predicts choice.

Thank you for catching this oversight. We have added cross-validated choice-prediction accuracy distributions to Figure 2B (~70% accuracy across models).

5) The main result of the paper (Figure 2) is based on effects averaged across different inactivation conditions (different epochs). However, I wonder if it makes sense to combine conditions like this. One, I wonder if this could hide areas that are involved during specific epochs of the task. The text states that "…aligned curves from different epochs were fairly consistent (Figure 2B)", but it is not clear how this is quantified and compared to what reference. Two, I wonder if this pooling would violate assumptions of statistical tests given data now comes from distinct sources, rather than being repeated observations.

The reviewer makes a good point. However, as detailed in our response to comment 1, we cannot be certain that those are legitimate differences between epochs, rather than just an artifact of inter-animal variability. Thus, we would rather make the more conservative choice of presenting them together, while still showing more granular per-condition analyses in the supplements. We also justify these choices more explicitly in the main text now. Finally, while our previous method of pooling data may have violated statistical assumptions, the fact that we are now fitting all data in the same model and explicitly modeling that variability as random effects should address that concern.

6) The analysis of calcium dynamics are based on the autoregressive component of the GLM model. This is counterintuitive because that component is not related to the stimulus or the task. If the claim is that evidence accumulation is related to the timescale of neural dynamics, shouldn't the analysis focus on the coefficients for E_δ (cumulative #R – #L towers), i.e. the component of the dynamics that encodes the stimulus?

We have now added the analysis suggested by the reviewer to Figure 4—figure supplement 1. Interestingly, we did not observe systematic timescale differences in E_δ, or in the responses locked to sensory-evidence pulses. This is now described in the Results section (line 315):

“We first wondered whether different timescales would be reflected in model coefficients related to sensory evidence. Interestingly, however, we did not observe any significant differences across areas in the time course of coefficients for contralateral tower stimuli or cumulative sensory evidence (Figure 4—figure supplement 1). Thus, we next focused our analysis on the auto-regressive coefficients of the model.”

At face value, this would suggest that the auto-regressive coefficients capture temporal components of neural dynamics that are not locked to any task events. Of course, the possibility remains that, despite our extensive parameterization of behavioral events, we failed to capture some task component that would display timescale differences across areas. We have added a discussion to acknowledge this possibility (line 332):

"Nevertheless, a caveat here is that the auto-regressive coefficients of the encoding model could conceivably be spuriously capturing variance attributable to other behavioral variables not included in the model. For example, our model parameterization implicitly assumes that evidence encoding would be linearly related to the side difference in the number of towers. Although this is a common assumption in evidence-accumulation models (e.g., Bogacz et al., 2006; Brunton et al., 2013), it could not apply to our case."

7) In a couple of places in the text, I feel the claims should be weakened as they go beyond the data. For example,a. Intro: "… provide the first casual demonstration that this hierarchy [of timescale] is important for cognitive behavior." A similar statement is in the 2nd paragraph of discussion. I suggest changing the framing. The experiments do not manipulate the timescale of cortical regions. The relationship with the observed behavioral deficit is correlative.

Following the reviewer's suggestion, we have removed these claims altogether.

b. Page 11, "This suggests that signals from the different dorsal cortical areas could be combined by downstream regions in a near-linear fashion. Candidate regions include … " The following paragraph is perhaps more suitable for discussion since the experiments do not probe subcortical regions. Also see comment 8 below. The effects of combined-area inactivation in fact appear to be qualitatively different from the average of single area silencing.

This has been moved to the discussion as suggested. Also please note that, as we elaborate in the response to comment 8 below, our new analysis methods did reveal significant differences between simultaneous silencing and single-area averages. The discussion has been updated accordingly.

c. Page 13, "…the different intrinsic timescales across the cortex support evidence integration over time windows of different durations." For the same reason as in comment (a) above, I suggest rephrasing or removing this framing.

We have rewritten this sentence as: “our findings could suggest that the different intrinsic timescales across the cortex are important for evidence-accumulation computations.” Note that this now also immediately follows our discussion on the caveats about the auto-regressive coefficients (see response to comment 6).

d. Abstract and intro, "inactivation of different areas primarily affected the evidence-accumulation per se, rather than other decision-related process". It seems the results do not examine other decision-related process besides the weighting of sensory evidence.

Following the reviewer’s suggestion, we have changed this framing throughout. While we do believe that our revised analyses indicate disruptions to the accumulation process (see response to comments 2 and 3 above), we agree that we did not fully examine other alternatives, so we removed our claims that evidence accumulation is the only affected process.

e. The text claims the spatial resolution of inactivation is 1.5-2mm. This is somewhat misleading. In Figure S2 of Pinto 2019, 60% of neurons are silenced at this light intensity at 2mm from light center. This broad inactivation is also consistent with the characterization from the Svoboda lab (Li et al., eLife 2019), which suggests that the spread of inactivation at 6 mW extends well beyond 2 mm in radius.

The original estimate referred to full inactivation, but the reviewer is of course correct that we still saw partial inactivation at 2 mm. We have therefore replaced these statements in the text by "≥ 2 mm" (lines 146, 517).

8) In Figure 2-Supp 3, the effects of posterior vs frontal cortex inactivation do not appear to be very different from each other. This is somewhat different from the averages of single area effects. In general, the statistical tests in the paper do not directly compare the effects of posterior cortex inactivation vs. frontal cortex inactivation. A more appropriate test for the key conclusion should be an interaction of y-position dependence with cortical regions.

The reviewer is correct that simultaneous inactivation appears to yield qualitatively different results than individual area averages. Our previous statistical procedures did not capture significant differences given their low power. However, our new analysis indeed revealed those differences to be significant (Figure 3—figure supplement 1). Despite this, it remains the case that frontal inactivation caused larger deficits on longer timescales, either when comparing single-area averages (Figure 2C) or simultaneous inactivation (Figure 3). We have expanded the description of these findings in the Results section, copied below for the reviewer's convenience (line 232):

"(…) Indeed, frontal and posterior areas differed significantly in terms of the magnitude and time course of evidence-weighting deficits induced by their inactivation (Figure 2C, 2-way repeated measure ANOVA with factors time bin and area group; F(time)_5,15_ = 3.09, p(time) = 0.047, F(area)_1,3_ = 33.93, p(area) = 0.010, F(interaction)_5,15_ = 3.60, p(interaction) = 0.025).

To further explore the different contributions of posterior and frontal cortical areas to the decision-making process, we next analyzed the effect of inactivating these two groups of areas simultaneously, using the same mixed-effects modeling approach as above. Compatible with our previous analysis, we found significant differences in how these two manipulations impacted the use of sensory evidence (Figure 3). In particular, compared to posterior areas, frontal inactivation resulted in a significantly larger decrease in the use of sensory evidence occurring long before laser onset (1.0 – 1.5 s, p = 0.006, z-test). Moreover, it led to decreases in the use of sensory evidence occurring after inactivation (p < 0.001, z-test), lending further support for a role of these regions in post-accumulation processes.

Finally, we wondered whether evidence information from different areas is evenly combined, at least from a behavioral standpoint. To do this, we compared the effects of simultaneously inactivating all frontal or posterior areas to that expected by an even combination of the effects of inactivating areas individually (i.e. their average). Both posterior and frontal significantly deviated from the even-combination prediction (Figure 3—figure supplement 1, p < 0.05, z-test). This could suggest that signals from the different dorsal cortical areas are combined with different weights towards a final decision."

9) The explanation of power analysis is not very clear (page 26-27). How are the control trials subsampled at different number of inactivation trials? What does it mean to bootstrap all the inactivation conditions together? At what effect size is n=250 sufficient to detect the effect?

We regret that this section was not written clearly enough. We have rewritten it to clarify the reviewer’s questions (line 672):

“To estimate statistical power, we performed a bootstrapping-based power analysis based on the one described by Guo et al. (Guo et al., 2014). We randomly subsampled the full dataset containing all inactivation conditions. In each subsample, we selected different numbers of inactivation trials regardless of area-epoch combination (50 < n < 1000, in steps of 25), and added a random subset of control trials such that the relative proportion of control to laser trials was preserved in the subsampled dataset. We then ran the bootstrapping procedure described above to compute laser-induced changes in overall performance combined across all inactivation conditions, extracting p-values for each of the values of n subsamples. We repeated this procedure ten times. Power was defined as the minimum number of trials required to observe p < 0.05 at the empirical effect size pooled across conditions, as defined by the first n where the 2 x SEM across the 10 repeats is below 0.05. We obtained an aggregate power of n = 250.”

10) The non-monotonic effect of cluster 3 (V1 and RSC) in Figure 2c is counterintuitive. The effect seems to be present in several individual conditions in Figure 1-Supp 2. However, other conditions don't show this (e.g. delay epoch inactivation). The text states that the effect is potentially compatible with findings that multiple timescales exist in a single region. Please explain this notion more clearly and how it could lead to no deficit for recent stimulus information but deficits for distant stimulus memory.

We share the reviewer's puzzlement about these findings. However, they have remained true even after we accounted for inter-subject and inter-condition variability. Following the reviewer's suggestion, we have expanded the discussion of the findings, including possible technical artifacts leading to them (Discussion, line 375):

"This could be in part due to technical limitations of the experiments. First, the laser powers we used result in large inactivation spreads, potentially encompassing neighboring regions. Moreover, local inactivation could result in changes in the activity of interconnected regions (Young et al., 2000), a possibility that should be evaluated in future studies using simultaneous inactivation and large-scale recordings across the dorsal cortex. At face value, however, the findings could be a reflection of the fact that diverse timescales exist at the level of individual neurons within each region (Bernacchia et al., 2011; Cavanagh et al., 2020; Scott et al., 2017; Spitmaan et al., 2020; Wasmuht et al., 2018). For example, inactivating an area with multimodal distributions of intrinsic timescales across its neurons could conceivably result in non-monotonic effects of inactivation."

11) Mice speed up during photostimulation in nearly all conditions (Figure 2-Supp 1). Are mice responding to the light? Ideally, a negative control could be included to show there are no non-specific effects of photostimulation when analyzed in the logistic regression. This could be done by photostimulation in GFP mice or by inactivation a cortical region not involved in the behavior.

We performed the suggested controls in the context of a recent paper, in which we used identical photostimulation parameters (Pinto et al., 2019). We did not observe the significant effects on running speed for any of the locations tested here when we performed the experiments in mice not expressing channelrhodopsin (and only a minor effect in 1/29 tested locations). We have added a statement to this effect when describing these results (line 151):

“Importantly, we have previously shown that these effects are specific to mice expressing ChR2, ruling out a non-specific light effect (Pinto et al., 2019).”

Reviewer #3 (Recommendations for the authors):Related to the above comment on aggregating data across mice, the presentation of the data would be more transparent if mouse-by-mouse results were shown, where possible (like they are in Figure 1B,C; Figure 1-table S1 is also helpful). For example, symbols for individual mice could be shown in Figure 1E instead of (or in addition to) the mean across mice. Presumably change in performance was calculated within mice and then averaged, rather than averaging laser on and laser off performance across mice and then taking the difference between the two. But the description ("inactivation-induced change in overall % correct performance for each inactivation epoch, for data combined across mice", line 119) could apply to either analysis.

We have now added individual mouse data where applicable (Figures 1E, 2A, 3A, 4G, Figure 1—figure supplement 3, Figure 2—figure supplement 1, Figure 2—figure supplement 2, Figure 4—figure supplement 1B,D). However, performance was in fact calculated on data pooled across mice and averaged across bootstrapping iterations (see Materials and methods) to account for different numbers of trials across mice. Thus, the displayed averages are closer to trial-count-weighted averages across mice. We now realize that the wording was unclear, and have changed “*combined* across mice” to “*pooled* across mice” where relevant.